# Cd59 and inflammation regulate Schwann cell development

Ashtyn T Wiltbank[1,2], Emma R Steinson[3], Stacey J Criswell[4], Melanie Piller[3], Sarah Kucenas[1,2,3]*

[1]Neuroscience Graduate Program, University of Virginia, Charlottesville, United States; [2]Program in Fundamental Neuroscience, University of Virginia, Charlottesville, United States; [3]Department of Biology, University of Virginia, Charlottesville, United States; [4]Department of Cell Biology, University of Virginia, Charlottesville, United States

**Abstract** Efficient neurotransmission is essential for organism survival and is enhanced by myelination. However, the genes that regulate myelin and myelinating glial cell development have not been fully characterized. Data from our lab and others demonstrates that *cd59*, which encodes for a small GPI-anchored glycoprotein, is highly expressed in developing zebrafish, rodent, and human oligodendrocytes (OLs) and Schwann cells (SCs), and that patients with CD59 dysfunction develop neurological dysfunction during early childhood. Yet, the function of Cd59 in the developing nervous system is currently undefined. In this study, we demonstrate that *cd59* is expressed in a subset of developing SCs. Using *cd59* mutant zebrafish, we show that developing SCs proliferate excessively and nerves may have reduced myelin volume, altered myelin ultrastructure, and perturbed node of Ranvier assembly. Finally, we demonstrate that complement activity is elevated in *cd59* mutants and that inhibiting inflammation restores SC proliferation, myelin volume, and nodes of Ranvier to wildtype levels. Together, this work identifies Cd59 and developmental inflammation as key players in myelinating glial cell development, highlighting the collaboration between glia and the innate immune system to ensure normal neural development.

*For correspondence:
sk4ub@virginia.edu

Competing interest: The authors declare that no competing interests exist.

## Editor's evaluation

This work investigates the function of the small GPI-anchored protein Cd59, a protein known to suppress complement-mediated inflammation, in Schwann cell development and myelination, using zebrafish as a model system. This article will be of interest to developmental biologists, glial biologists, and immunologists as it suggests the interesting and novel findings that Cd59 regulates Schwann cell development, mainly by modulating Schwann cell proliferation.

## Introduction

Myelin is a highly specialized, lipid-rich membrane that insulates axons in the nervous system to enhance neurotransmission (*Ritchie, 1982*) while preserving axon health (*Stadelmann et al., 2019*). Myelin is produced and maintained by myelinating glial cells such as motor exit point glia (*Fontenas and Kucenas, 2021*; *Fontenas and Kucenas, 2018*; *Smith et al., 2014*) and myelinating Schwann cells (SCs; *Figure 1A*; *Ackerman and Monk, 2016*; *BEN GEREN, 1954*; *Jessen and Mirsky, 2005*) in the peripheral nervous system (PNS) as well as oligodendrocytes (OLs) in the central nervous system (CNS) (*Figure 1B*; *Ackerman and Monk, 2016*; *Maturana, 1960*; *Nave and Werner, 2014*; *Peters, 1960a*; *Peters, 1960b*). Normal myelin and myelinating glial cell development facilitate an efficient and effective nervous system, ensuring precise motor, sensory, and cognitive function (*Almeida and*

**eLife digest** The nervous system of vertebrates is made of up of complex networks of nerve cells that send signals to one another. In addition to these cells, there are a number of supporting cells that help nerves carry out their role. Schwann cells, for example, help nerve cells to transmit information faster by wrapping their long extensions in a fatty membrane called myelin. When myelin is not produced properly, this can disturb the signals between nerve cells, leading to neurological defects.

Schwann cells mature simultaneously with nerve cells in the embryo. However, it was not fully understood how Schwann cells generate myelin during development. To investigate, Wiltbank et al. studied the embryos of zebrafish, which, unlike other vertebrates, are transparent and develop outside of the womb. These qualities make it easier to observe how cells in the nervous system grow in real-time using a microscope.

First, the team analyzed genetic data collected from the embryo of zebrafish and mice to search for genes that are highly abundant in Schwann cells during development. This led to the discovery of a gene called *cd59*, which codes for a protein that also interacts with the immune system. To find out whether Schwann cells rely on *cd59*, Wiltbank et al. deleted the *cd59* gene in zebrafish embryos. Without *cd59*, the Schwann cells produced too many copies of themselves and this, in turn, impaired the appropriate production of myelin.

Since the protein encoded by *cd59* normally balances inflammation levels, it was possible that losing this gene overactivated the immune system in the zebrafish embryos. In support of this hypothesis, when the *cd59* mutant embryos were treated with an anti-inflammatory drug, this corrected Schwann cell overproduction and restored myelin formation.

Taken together, these findings reveal how inflammation helps determine the number of Schwann cells produced during development, and that *cd59* prevents this process from getting carried away. This suggests that the nervous system and immune system may work together to build the nervous system. In the future, it will be interesting to investigate whether *cd59* acts in a similar way during the development of the human nervous system.

*Lyons, 2017*; *Berger et al., 2006*; *Stadelmann et al., 2019*; *Wei et al., 2019*). When this process is impaired, as seen with inherited disorders that cause abnormal myelination, patients present with neurological dysfunction that often leads to severe physical and intellectual disabilities (*Berger et al., 2006*; *Stadelmann et al., 2019*; *van der Knaap and Bugiani, 2017*; *Wei et al., 2019*).

Despite the importance of myelin, we still lack a complete understanding of myelin and myelinating glial cell development. With the recent boon of transcriptomic and proteomic analyses, many genes and proteins have been highlighted as differentially expressed during myelinating glial cell development, yet their precise functions remain unknown. For example, *cd59*, a gene that encodes for a small, glycosylphosphotidylinositol (GPI)-anchored glycoprotein, is a particularly interesting candidate for exploration. This gene features in several RNA sequencing (RNAseq) and proteomic analyses of zebrafish and rodent myelinating glial cells (*Gerber et al., 2021*; *Howard et al., 2021*; *Marisca et al., 2020*; *Marques et al., 2018*; *Marques et al., 2016*; *Piller et al., 2021*; *Saunders et al., 2019*; *Siems et al., 2021*; *Wolbert et al., 2020*; *Zhu et al., 2019*), all of which demonstrate high expression of *cd59* in developing oligodendrocyte lineage cells (OLCs) and SCs. *cd59* continues to be expressed in juvenile and adult zebrafish (*Saunders et al., 2019*; *Siems et al., 2021*; *Sun et al., 2013*) and becomes the fourth most abundant CNS myelin protein in adult zebrafish (*Siems et al., 2021*). Phylogenetic analysis demonstrates that Cd59 is conserved within vertebrate evolution (*Siems et al., 2021*), and accordingly, CD59 protein is also found in developing and adult human myelin (*Erne et al., 2002*; *Koski et al., 2002*; *Scolding et al., 1998*; *Zajicek et al., 1995*). Collectively, these data indicate that *cd59* is a key component of the genetic program that orchestrates myelinating glial cell development. However, despite this knowledge of Cd59 expression, little is known about the function of Cd59 in the developing nervous system.

Outside of the nervous system, Cd59 is best known as a complement inhibitory protein. Complement is a molecular cascade within the innate immune system that aids in cell lysis of invading pathogens or aberrant cells in the body (*Merle et al., 2015*). This process lacks specificity and requires inhibitory proteins, such as Cd59, to protect healthy cells from lytic death (*Davies et al., 1989*). Cd59,

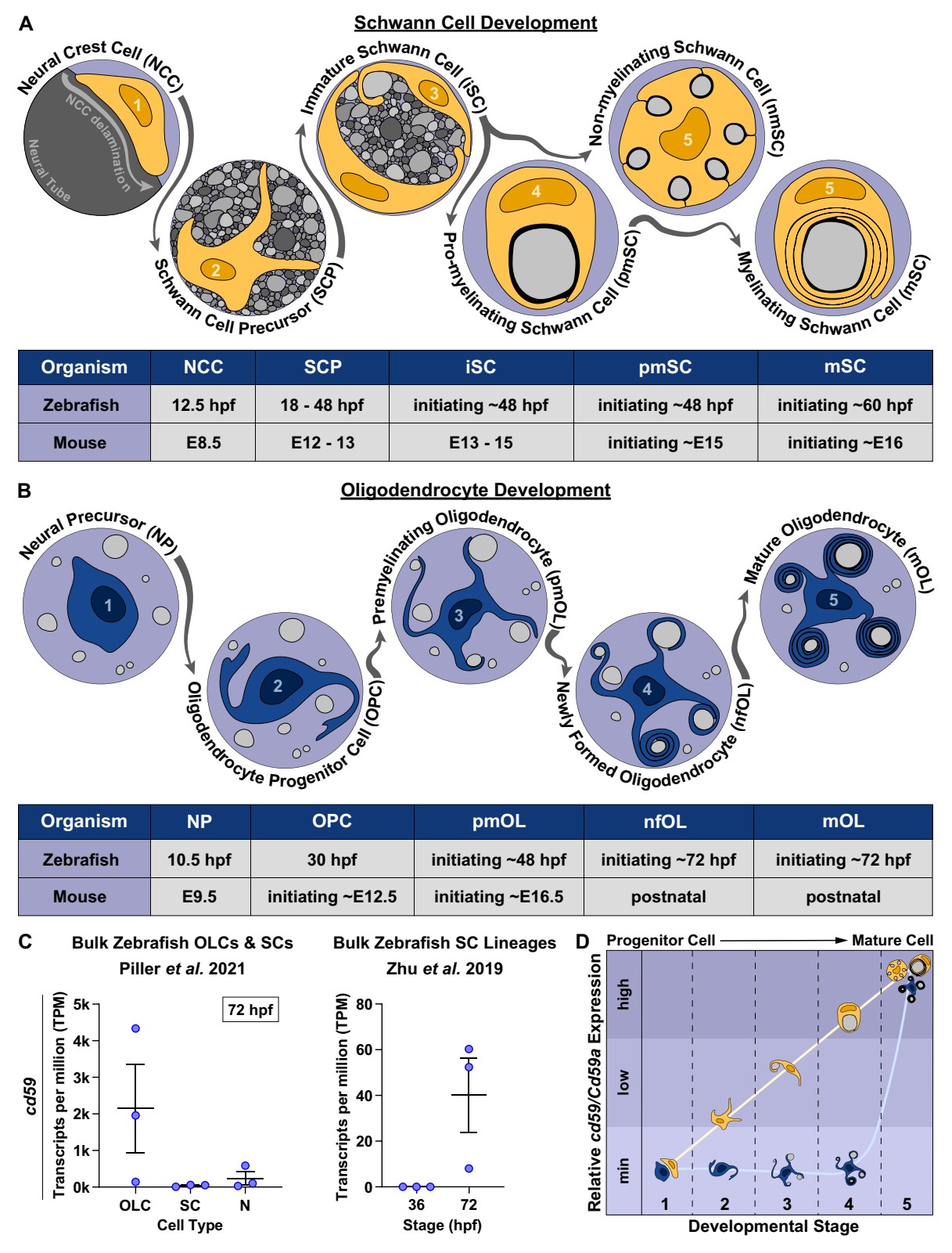

**Figure 1.** *cd59* is expressed in myelinating glial cells during nervous system development. (**A**) Timeline of Schwann cell (SC) (orange) development (top panel). SC developmental stages for zebrafish (hours post fertilization [hpf]) and mice (embryonic day [E]) are indicated in the bottom panel. (**B**) Timeline of oligodendrocyte (OL) (blue) development. OL developmental stages for zebrafish (hpf) and mice (E and postnatal day) are indicated in the bottom panel. (**C**) Scatter plot of *cd59* expression (TPM) in oligodendrocyte lineage cells (OLCs), SCs, and neurons (N) at 72 hpf (left; mean ± SEM: OLC: 2145.1

*Figure 1 continued on next page*

*Figure 1 continued*

± 1215.1; SC: 40.1 ± 16.3; N: 240.5 ± 173.3; dot = replicate) as well as SCs at 36 and 72 hpf (right; mean ± SEM: 36 hpf: 0.0 ± 0.0, 72 hpf: 40.2 ± 16.3; dot = replicate). (**D**) Schematic of the relative *cd59/Cd59a* expression in developing SCs (orange) and OLs (blue) determined from RNAseq analysis in *Figure 1—figure supplement 1*. Developmental stage numbers correspond with stages indicated in (**A**) and (**B**). Artwork created by Ashtyn T. Wiltbank with Illustrator (Adobe) based on previous schematics and electron micrographs published in *Ackerman and Monk, 2016*; *Cunningham and Monk, 2018*; *Jessen and Mirsky, 2005*.

The online version of this article includes the following source data and figure supplement(s) for figure 1:

**Source data 1.** Source data for *cd59* bulk, RNAseq expression depicted in *Figure 1C*.

**Figure supplement 1.** Myelinating glial cell *cd59/Cd59a* expression from bulk and scRNAseq.

specifically, acts at the end of the complement cascade where it binds to complement proteins 8 and 9 (C8 and C9, respectively) to prevent the polymerization of C9 and the subsequent formation of lytic pores (also known as membrane attack complexes [MACs] or terminal complement complexes [TCCs]) in healthy cell membranes (*Meri et al., 1990*; *Ninomiya and Sims, 1992*; *Rollins et al., 1991*). This interaction is important throughout the adult body, including the nervous system, where CD59a is neuroprotective in models of multiple sclerosis (*Mead et al., 2004*) and neuromyelitis optica (*Yao and Verkman, 2017*). Beyond complement-inhibitory functions, Cd59 can also facilitate vesicle signaling in insulin-producing pancreatic cells (*Golec et al., 2019*; *Krus et al., 2014*), suppress cell proliferation in T cells responding to a viral infection (*Longhi et al., 2005*) or smooth muscle cells in models of atherosclerosis (*Li et al., 2013*), and orchestrate proximal-distal cell identity during limb regeneration (*Echeverri and Tanaka, 2005*). Like complement inhibition, these processes are critical to normal nervous system function (*Almeida et al., 2021*; *Baron and Hoekstra, 2010*; *Reiter and Bongarzone, 2020*; *White and Krämer-Albers, 2014*). However, it is unclear exactly what role Cd59 is playing in myelinating glial cell development.

Though the precise function is unknown, it is clear that CD59 does impact human nervous system development. Patients with CD59 dysfunction, such as those with congenital CD59 deficiency (*Haliloglu et al., 2015*; *Höchsmann and Schrezenmeier, 2015*; *Karbian et al., 2018*; *Solmaz et al., 2020*) or germline paroxysmal nocturnal hemoglobinuria (*Johnston et al., 2012*), present with poly-neuropathies during infancy and continue to have nervous system dysfunction throughout their lives. Intriguingly, these neurological symptoms persist even with complement inhibition, the most common treatment for congenital CD59 deficiency (*Höchsmann and Schrezenmeier, 2015*). Together, these observations further indicate that CD59 has an additional role in the developing nervous system and requires further investigation.

Here, utilizing the zebrafish model, we examined the role of Cd59 in the developing nervous system. In this study, we found that *cd59* is highly expressed in a subset of developing SCs in addition to mature OLs and myelinating and nonmyelinating SCs. We chose to focus on SCs and found that *cd59* mutant zebrafish have excessive SC proliferation. These mutants also have impaired myelin and node of Ranvier development. Finally, we demonstrate that complement activity increases in *cd59* mutants and that unregulated inflammation contributes to SC overproliferation and mutant larvae may also have aberrant myelin and node of Ranvier formation. Overall, this data demonstrates that developmental inflammation stimulates SC proliferation and that this process is balanced by Cd59 to ensure normal SC and myelinated nerve development.

## Results

### *Cd59* is expressed in myelinating glial cells during nervous system development

Myelinating glial cell development is a complex process (*Figure 1A and B*) orchestrated by a genetic program that is not yet fully elucidated. To identify unexplored genes that may impact myelinating glial cell or myelin development, we evaluated bulk and single-cell (sc) RNAseq datasets that assessed myelinating glial cells during nervous system development (*Gerber et al., 2021*; *Howard et al., 2021*; *Marisca et al., 2020*; *Marques et al., 2018*; *Marques et al., 2016*; *Piller et al., 2021*; *Saunders et al., 2019*; *Wolbert et al., 2020*; *Zhu et al., 2019*). Across multiple datasets, *cd59* (zebrafish) or *Cd59a* (mouse), which encodes for a small, GPI-anchored glycoprotein that has no known function in

the developing nervous system, is expressed in SCs and OLCs (*Figure 1C*, *Figure 1—figure supplement 1*).

When looking at SCs, data from developing zebrafish and mice indicated that *cd59/Cd59a*, respectively, is minimally expressed in neural crest cells (NCCs; the multipotent progenitors that give rise to SCs; *Figure 1—figure supplement 1A*; *Howard et al., 2021*; *Zhu et al., 2019*) but increases in expression in SC precursors (SCPs; *Figure 1—figure supplement 1B.1,B.2*; *Gerber et al., 2021*), immature SCs (iSCs; *Figure 1—figure supplement 1B.1,B.2*; *Gerber et al., 2021*), pro-myelinating SCs (pmSCs; *Figure 1—figure supplement 1B.1,B.2*; *Gerber et al., 2021*), and mature myelinating SCs (MSCs) and non-myelinating SCs (MSCs and NMSCs; *Figure 1C*, *Figure 1—figure supplement 1B.3,C.1,C.2*; *Gerber et al., 2021*; *Piller et al., 2021*; *Saunders et al., 2019*; *Wolbert et al., 2020*). This data indicates that Cd59 may be important during early stages of SC development as well as in mature MSCs and NMSCs (*Figure 1D*).

In contrast, bulk RNAseq of zebrafish oligodendrocyte progenitor cells (OPCs) and OLs indicated that *cd59* is expressed near the onset of myelination (72 hours post fertilization [hpf]; *Figure 1C*; *Piller et al., 2021*). scRNAseq (*Marisca et al., 2020*; *Marques et al., 2018*; *Marques et al., 2016*) and in situ hybridization (ISH; *Siems et al., 2021*) of zebrafish and mouse OLCs showed that *cd59/CD59a* is mostly expressed in mature OLs and not OPCs (*Figure 1—figure supplement 1D*). CD59 protein is also present within newly formed human myelin sheaths as is evident in immunostaining of third trimester fetal brains (*Zajicek et al., 1995*). Based on these findings, it is likely that Cd59 does not play a role in early stages of OLC development but could influence mature, myelinating OL function (*Figure 1D*).

Intrigued by this elevated expression of *cd59* in developing myelinating glial cells, we verified that *cd59* RNA was expressed in OLCs and SCs with chromogenic ISH (CISH) and fluorescent ISH (FISH) at 3 days post fertilization (dpf) (*Figure 2A–C*). Morphology, location, and co-expression with *sox10:megfp*, which is a marker for both OLCs and SCs, showed that *cd59* RNA is expressed in both cell types at 3 dpf (*Figure 2A–C*). Beyond expression in the heart and pancreas, *cd59* expression is largely confined to developing myelinating glial cells at this stage.

To study these cells in vivo, we created a transcriptional reporter zebrafish line for *cd59* (herein referred to as *cd59:tagrfp*). This line utilizes 5 kb of the *cd59* promoter as well as the first exon and intron, the latter containing an enhancer that is important for *cd59* expression (*Tone et al., 1999*), to drive cytoplasmic expression of TagRFP in *cd59*-expressing cells. With in vivo, confocal imaging, mosaic labeling with our *cd59:tagrfp* construct revealed labeling of OLs and SCs with myelin-like processes (*Figure 2D*), indicating that *cd59* is expressed in mature myelinating glial cells.

Bulk and scRNAseq data indicated that *cd59* is expressed during early stages of SC development whereas *cd59* expression was restricted to mature OLs. To verify this expression pattern, we looked at *cd59* RNA expression at earlier stages in the SC lineage. At 24 hpf, using CISH and FISH, we did not observe *cd59* expression in *sox10:megfp*-positive NCCs, though we did see it in the hypochord and floor plate of the spinal cord at this stage (*Figure 2—figure supplement 1A and B*). At 36 and 48 hpf, when NCCs begin to develop into SCPs and iSCs, we detected sporadic expression of *cd59* with FISH in *sox10:megfp*-positive cells along the posterior lateral line nerve (pLLN) in the PNS (*Figure 2—figure supplement 1C and D*). By 72 hpf, we observed robust *cd59* expression within *sox10:megfp*-positive SCs along the pLLN (*Figure 2—figure supplement 1C and D*). These data demonstrate that *cd59* is expressed in developing SCs, confirming the expression patterns seen in the RNAseq datasets we analyzed (*Figure 1—figure supplement 1*).

From our ISH data, we noticed that *cd59* was not expressed in every SC. Notably, SCs associated with spinal motor nerves did not express *cd59* (*Figure 2A–C*). Rather, *cd59* expression was largely confined to SCs along the pLLN (*Figure 2A–C*). To investigate this expression further, we utilized our stable *cd59:tagrfp* line. This transgenic line provides clear, in vivo labeling of c*d59*-expressing pLLN SCs, which we confirmed using FISH with a *cd59* probe in *cd59:tagrfp*-positive SCs (*Figure 2—figure supplement 2A,B, B'*). To verify that *cd59:tagrfp*-positive SCs expressed canonical SC markers, including *SRY-box transcription factor 10* (*sox10*), *erb-b2 receptor tyrosine kinase 3b* (*erbb3b*), and *myelin basic protein a* (*mbpa*), we examined pLLN SCs in transgenic and gene trap larvae co-expressing these markers with in vivo confocal imaging. As expected, *cd59:tagrfp* expression was observed in *sox10*, *erbb3b,* and *mbp*-positive SCs at 72 and 7 dpf (*Figure 2—figure supplement 2C–E*), confirming that *cd59:tagrfp*-positive SCs express other known SC genes.

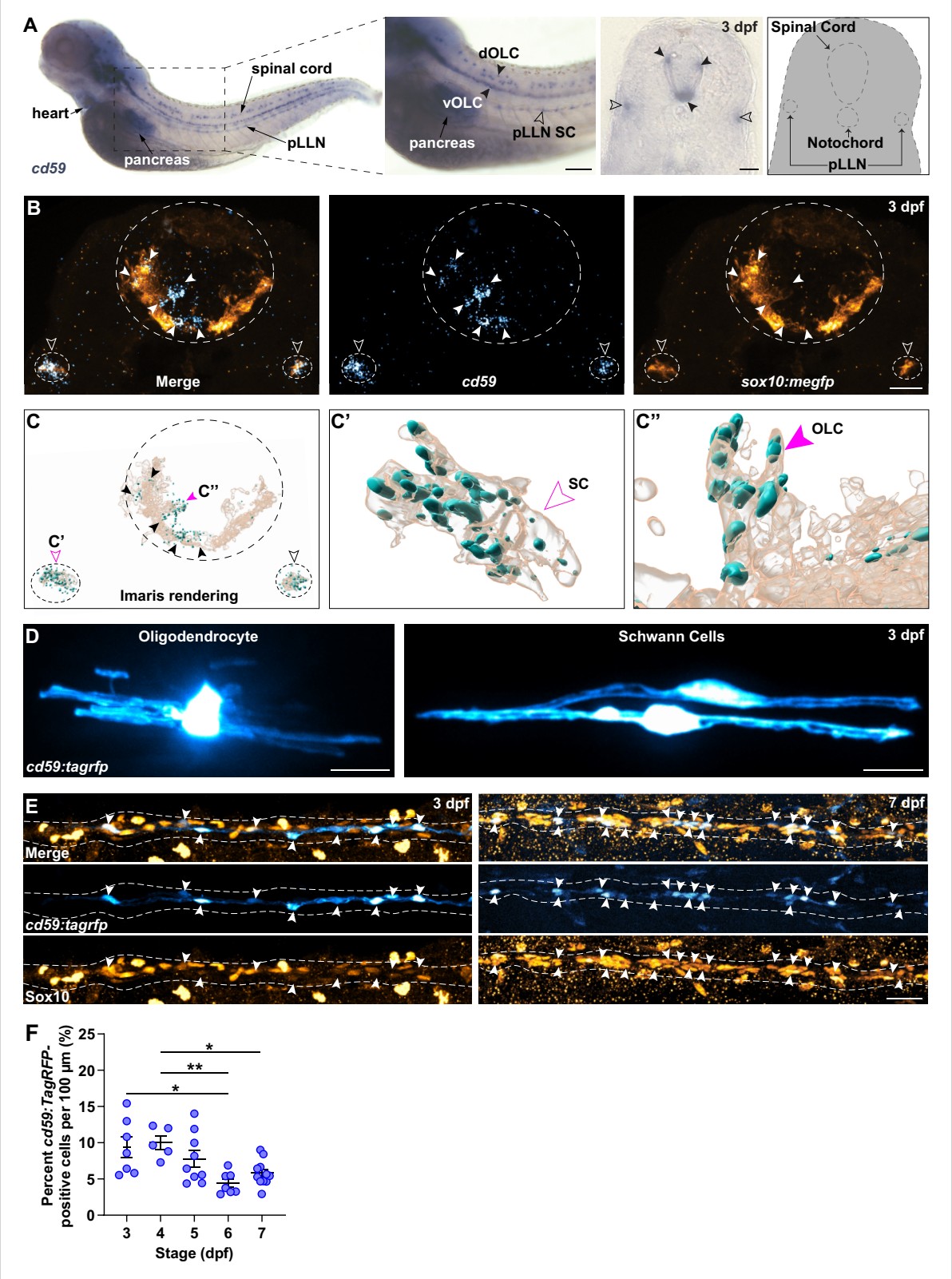

**Figure 2.** *cd59* is expressed in Schwann cells (SCs) and oligodendrocytes (OLs). (**A**) Bright-field images of whole-mount chromogenic in situ hybridization (CISH) showing *cd59* expression (purple) in the heart, pancreas, dorsal and ventral oligodendrocyte lineage cells (OLCs) (dOLCs and vOLCs, respectively; filled arrows), and posterior lateral line nerve (pLLN) SCs (empty arrows) at 3 days post fertilization (dpf). Schematic (right panel) indicates location of spinal cord, notochord, and pLLN in transverse section. (**B**) Fluorescent in situ hybridization (FISH) (RNAscope; ACD) showing *cd59*

*Figure 2 continued*

expression (cyan) in *sox10:megfp*-positive, pLLN SCs (*cd59*-positive orange cells indicated by empty arrows), and spinal cord OLs (*cd59*-positive orange cells indicated by filled arrows) at 3 dpf in transverse sections (top row). Representative image (top row) displays one SC on each pLLN (left and right empty arrows) as well as multiple OLs in the spinal cord (filled arrows). Images were acquired with confocal imaging. (**C**) Imaris renderings of the confocal images shown in (**B**), including the full image (left panel). From the full image (left panel, **C**), a single SC (indicated with the open magenta arrow and enlarged in the middle panel, **C'**) and a single OLC (indicated with the filled magenta arrow and enlarged in the right panel, **C''**) illustrate the *cd59* puncta localized within the myelinating cells. (**D**) Mosaic labeling showing a *cd59:tagrfp*-positive OL in the spinal cord (left) and two SCs on the pLLN (right) at 3 dpf. (**E**) Immunofluorescence (IF) showing *cd59:tagrfp* expression (cyan) in Sox10-positive SCs (orange) along the pLLN at 3 and 7 dpf (left and right panels, respectively). Double-positive cells are indicated with white arrows. White dashed lines outline the pLLN. Sox10-positive pigment cells outside of the dashed lines were not included in the analysis. (**F**) Scatter plot of percent *cd59:tagrfp*-positive SCs on the pLLN from 3 to 7 dpf (mean ± SEM: 3 dpf: 9.4 ± 1.5; 4 dpf: 10.0 ± 1.0; 5 dpf: 7.8 ± 1.2; 6 dpf: 4.4 ± 0.6; 7 dpf: 5.8± 0 .6; p-values: 3 vs. 6 dpf: p=0.0126, 4 vs. 6 dpf: p=0.0095, 4 vs. 7 dpf: *P*p0.0477; dot = 1 fish). Data collected from somites 11–13 (~320 µm) and normalized to units per 100 µm. These data were compared with a one-way ANOVA with Tukey's post-hoc test using GraphPad Prism. All fluorescent images were acquired with confocal imaging. Scale bars: (**A**) lateral view, 100 µm; transverse section, 25 µm; (**B, D**) 10 µm; (**E**) 25 µm. Artwork created by Ashtyn T. Wiltbank with Illustrator (Adobe).

The online version of this article includes the following source data and figure supplement(s) for figure 2:

**Source data 1.** Source data for the quantification of percent *cd59:tagrfp*-positive cells per 100 µm depicted in *Figure 2E*.

**Figure supplement 1.** *cd59* is expressed in the floorplate, hypochord, and developing Schwann cell (SCs).

**Figure supplement 2.** *cd59:tagrfp*-positive Schwann cells (SCs) express canonical SC markers.

To determine whether *cd59* was expressed in all pLLN SCs, we used immunofluorescence (IF) to co-label *cd59:tagrfp*-positive SCs with an antibody against Sox10, which labels all SCs along the pLLN. Strikingly, when looking at SCs from 3 to 7 dpf, *cd59:tagrfp* was only expressed in a subset of pLLN SCs (average of 4.4 ± 1.5% to 9.4 ± 3.8% of pLLN SCs; *Figure 2E and F*). We also noted that *cd59* expression was only expressed in a subset of OLs (*Figure 2B, C, C''*) and was completely lacking in MEP glia. Taken together, these data demonstrate that *cd59* expression is heterogeneous in developing myelinating glial cells.

## Generation of *cd59* mutant zebrafish with CRISPR/Cas9 genome editing

Our findings demonstrate that *cd59* is expressed in a subset of developing SCs. However, it is unclear how Cd59 influences SC development. In other cells, Cd59 has many different functions, including preventing complement-dependent cell lysis (*Davies et al., 1989*), facilitating vesicle-dependent signaling (*Golec et al., 2019*; *Krus et al., 2014*), suppressing cell proliferation (*Dashiell et al., 2000*; *Hila et al., 2001*; *Li et al., 2013*; *Longhi et al., 2005*), and orchestrating proximal-distal cell identity (*Echeverri and Tanaka, 2005*). There is little known about Cd59 in the developing nervous system, however.

To evaluate the function of Cd59 in SC development, we generated *cd59* mutant zebrafish with CRISPR/Cas9 genome editing. Targeting the end of the first coding exon (exon 2; *Figure 3A*), we recovered an allele with a 15 bp deletion that occurred across the exon 2–intron 2 splice site, generating a splice mutation (herein referred to as mutant *cd59^uva48^*; *Figure 3B*). RT-PCR and Sanger sequencing analysis of mutant *cd59^uva48^* transcripts revealed that multiple splice variants were produced compared to wildtype *cd59*, which produced a single transcript (*Figure 3—figure supplement 1A and E*). Analysis of the mutant transcripts indicated that an early stop codon was generated in the N-terminal signal sequence, producing a severely truncated protein (10–48 AA compared to the 119 AA wildtype protein; *Figure 3—figure supplement 1E and G*). Furthermore, the predicted proteins lacked the Ly6/uPAR domain and GPI-anchor signal sequence (*Figure 3—figure supplement 1E and G*). Therefore, the mutant Cd59 protein was expected to be nonfunctional. We also recovered an additional *cd59* mutant allele, *cd59^uva47^*, which was produced with the same single-guide RNA (sgRNA). This mutant had a 6 bp deletion near the end of exon 2 that did not affect the splice site (*Figure 3—figure supplement 1B and D*) and therefore produced a single mutant transcript (*Figure 3—figure supplement 1B*). Sequencing of the mutant *cd59^uva47^* transcript predicted the absence of two amino acids (AA) in the N-terminal signal sequence, which would yield a slightly smaller protein (117 AA) compared to wildtype Cd59 (119 AA; *Figure 3—figure supplement 1E and G*). Though the protein was not severely truncated, we predicted that interfering with the N-terminal signal sequence would prevent protein trafficking to the endoplasmic reticulum and interrupt proper protein function. Collectively,

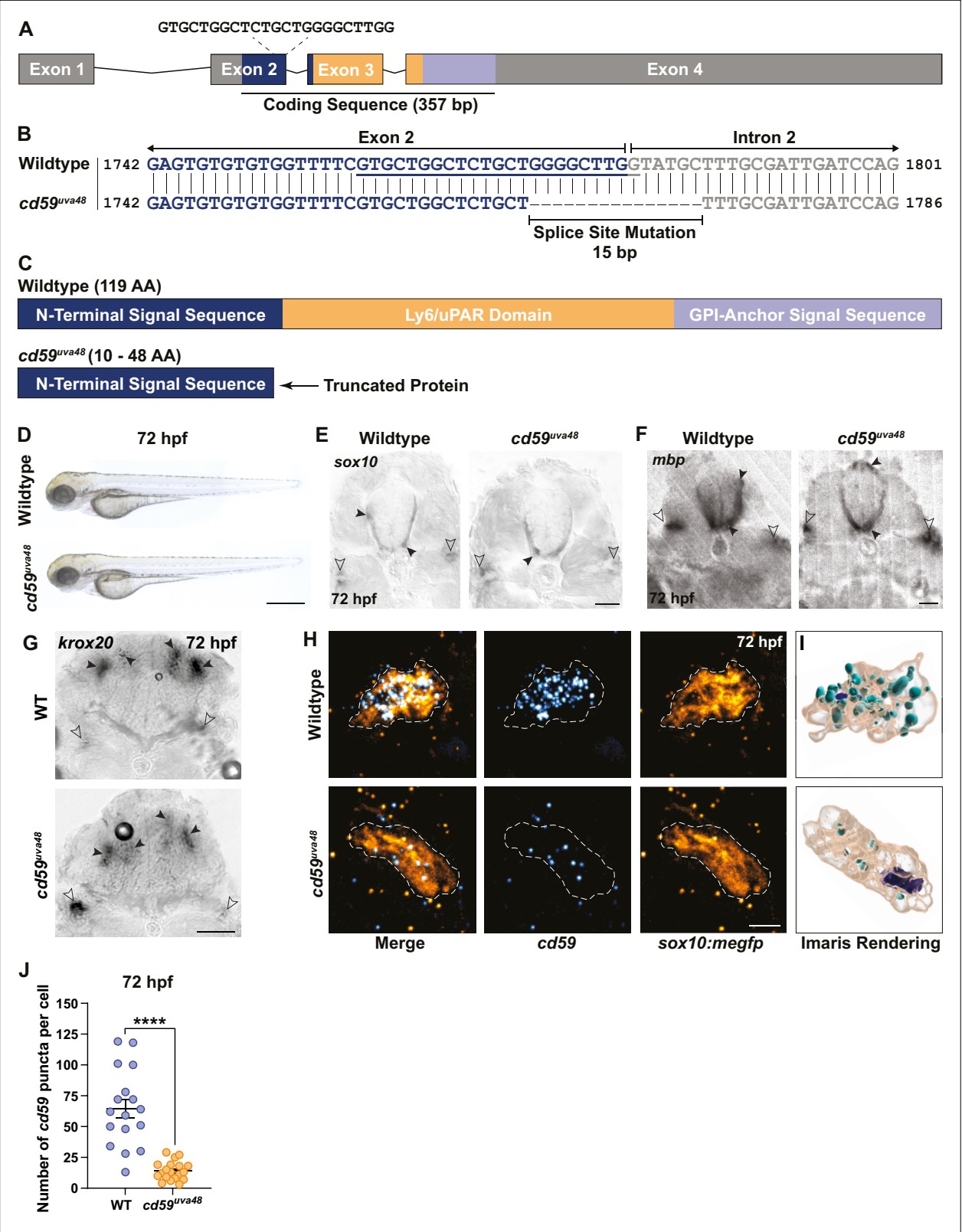

**Figure 3.** Generation of *cd59* mutant zebrafish with CRISPR/Cas9 genome editing. (**A**) Schematic of zebrafish *cd59* gene. The *cd59* coding sequence (357 bp; CDS) encodes the protein domains with the corresponding colors indicated in (**C**). The non-CDS is indicated in gray. The dashed lines indicate the sgRNA and the target sequence. (**B**) Genomic sequences for wildtype *cd59* and mutant *cd59*[uva48] showing a 15 bp deletion at the splice site between exon 2 (blue) and intron 2 (gray) of the *cd59* gene. (**C**) Schematic of Cd59 protein made in wildtype (119 AA; top panel) and *cd59*[uva48] mutant fish (10–48

*Figure 3 continued on next page*

*Figure 3 continued*

AA; bottom panel). (**D**) Bright-field images of wildtype and *cd59^uva48^* mutant larvae at 72 hours post fertilization (hpf) showing no anatomical defects as a result of *cd59* mutation. (**E**) Chromogenic in situ hybridization (CISH) showing *sox10*-positive (gray) oligodendrocytes (OLs) in the spinal cord (filled arrows) and Schwann cells (SCs) on the posterior lateral line nerve (pLLN) (empty arrows) at 72 hpf in transverse sections. (**F**) CISH showing *mbp*-positive (gray) OLs in the spinal cord (filled arrows) and SCs on the pLLN (empty arrows) at 72 hpf in transverse sections. (**G**) CISH showing *krox20*-positive (gray) SCs on the pLLN (empty arrows) and neurons in the brain (filled arrows) at 72 hpf in transverse sections. (**H**) Fluorescent in situ hybridization (FISH) (RNAscope; ACD) showing *cd59* expression (cyan) in *sox10:megfp*-positive SCs (orange) along the pLLN at 72 hpf. Representative images each display a transverse section (z projection of 20 μm) of single SC on the pLLN. (**I**) Imaris renderings show *cd59* puncta that are localized within each SC. (**J**) Scatter plot of the number of *cd59* RNA puncta in pLLN SCs (mean ± SEM: WT: 64.7 ± 7.6, *cd59^uva48^*: 14 ± 1.7; p<0.0001; dot = 1 cell; n = 7 fish per group). These data were compared with Student's *t*-test using GraphPad Prism. CISH and FISH images were acquired with confocal imaging. Scale bars: (**A**) 0.25 mm; (**E, F**) 25 μm; (**G**) 50 μm. Artwork created by Ashtyn T. Wiltbank with Illustrator (Adobe).

The online version of this article includes the following source data and figure supplement(s) for figure 3:

**Source data 1.** Source data for the quantification of the number of *cd59* puncta in Schwann cells (SCs) at 72 hours post fertilization (hpf) depicted in *Figure 3H*.

**Figure supplement 1.** Characterization of *cd59* mutant zebrafish.

**Figure supplement 1—source data 1.** Source data from gel electrophoresis of RT-PCR of *cd59^uva48^* mutant embryos.

**Figure supplement 1—source data 2.** Source data from gel electrophoresis of RT-PCR of *cd59^uva47^* mutant embryos.

mutant *cd59^uva48^* larvae have perturbed RNA splicing that results in premature termination codon (PTC) formation and probable protein truncation, and mutant *cd59^uva47^* larvae have a disrupted N-terminal sequence. Considering these results, both mutants were suspected to have impaired Cd59 function.

To begin our initial evaluation of these mutants, we sought to compare general developmental characteristics between *cd59* mutant larvae and their wildtype siblings. Both *cd59^uva48^* and *cd59^uva47^* homozygous adults were viable and produced embryos with no anatomical, behavioral, or reproductive defects (*Figure 3D*, *Figure 3—figure supplement 1C*). Next, we wanted to know if SCs were present in mutant *cd59^uva48^* larvae, which were utilized in the majority of our studies. Using CISH, we labeled wildtype and mutant *cd59^uva48^* larvae with RNA probes against *sox10*, *krox20* (also known as *early growth response 2b* or *egr2b),* and *mbp* at 72 hpf. Both wildtype and mutant larvae had *sox10*, *mbp,* and *krox20*-positive SCs along the pLLN (*Figure 3E–G*), indicating that Cd59 was not necessary for SC genesis or expression of genes necessary for SC maturation and myelination.

Knowing that SCs were present, we were then curious how *cd59* expression was affected in *cd59^uva48^* mutant SCs. According to our RT-PCR analysis, mutant *cd59^uva48^* embryos produced RNA transcripts with PTCs (*Figure 3—figure supplement 1E*). RNA transcripts with PTCs can sometimes undergo nonsense-mediated decay (NMD), a process by which normal and mutant gene expression is controlled through RNA degradation (*Bruno et al., 2011*; *Chang et al., 2007*; *Karam and Wilkinson, 2012*; *Lou et al., 2014*). To determine if mutant *cd59^uva48^* RNA transcripts underwent NMD, we used FISH to label wildtype and mutant *cd59^uva48^;sox10:megfp* larvae with a probe against *cd59* (*Figure 3H and I*). Quantification of *cd59* RNA puncta showed that *cd59* expression was significantly reduced in mutant SCs compared to wildtype SCs (*Figure 3J*), indicating that *cd59^uva48^* mutant RNA was being degraded through NMD and therefore could result in a loss of protein. Due to the lack of a zebrafish-specific antibody for Cd59, we were unable to determine if these changes in RNA were reflected at the protein level and therefore cannot confirm if we have produced a null allele for *cd59*. However, our findings indicate that *cd59^uva48^* mutants can be used as a loss-of-function model in which to study the role of Cd59 in developing SCs.

## Cd59 regulates developing SC proliferation

With our new *cd59* mutants, we began to explore the possible roles Cd59 could play in SC development. Previous studies demonstrate that CD59a can prevent overproliferation of T cells during a viral infection (*Longhi et al., 2005*). Similarly, CD59a is important for limiting deleterious smooth muscle cell proliferation during atherosclerosis (*Li et al., 2013*). Because we observed *cd59* expression during the proliferative phases of SC development, we hypothesized that Cd59 could be involved in developmental SC proliferation. To test our hypothesis, we labeled SCs with a Sox10 antibody and quantified the number of Sox10-positive SCs along the pLLN in wildtype and *cd59^uva48^* mutant larvae at various stages during SC development (36 hpf to 7 dpf) (*Figure 4A*). At all stages we investigated, there were significantly more SCs in mutant embryos and larvae compared to wildtype siblings (*Figure 4A*). We

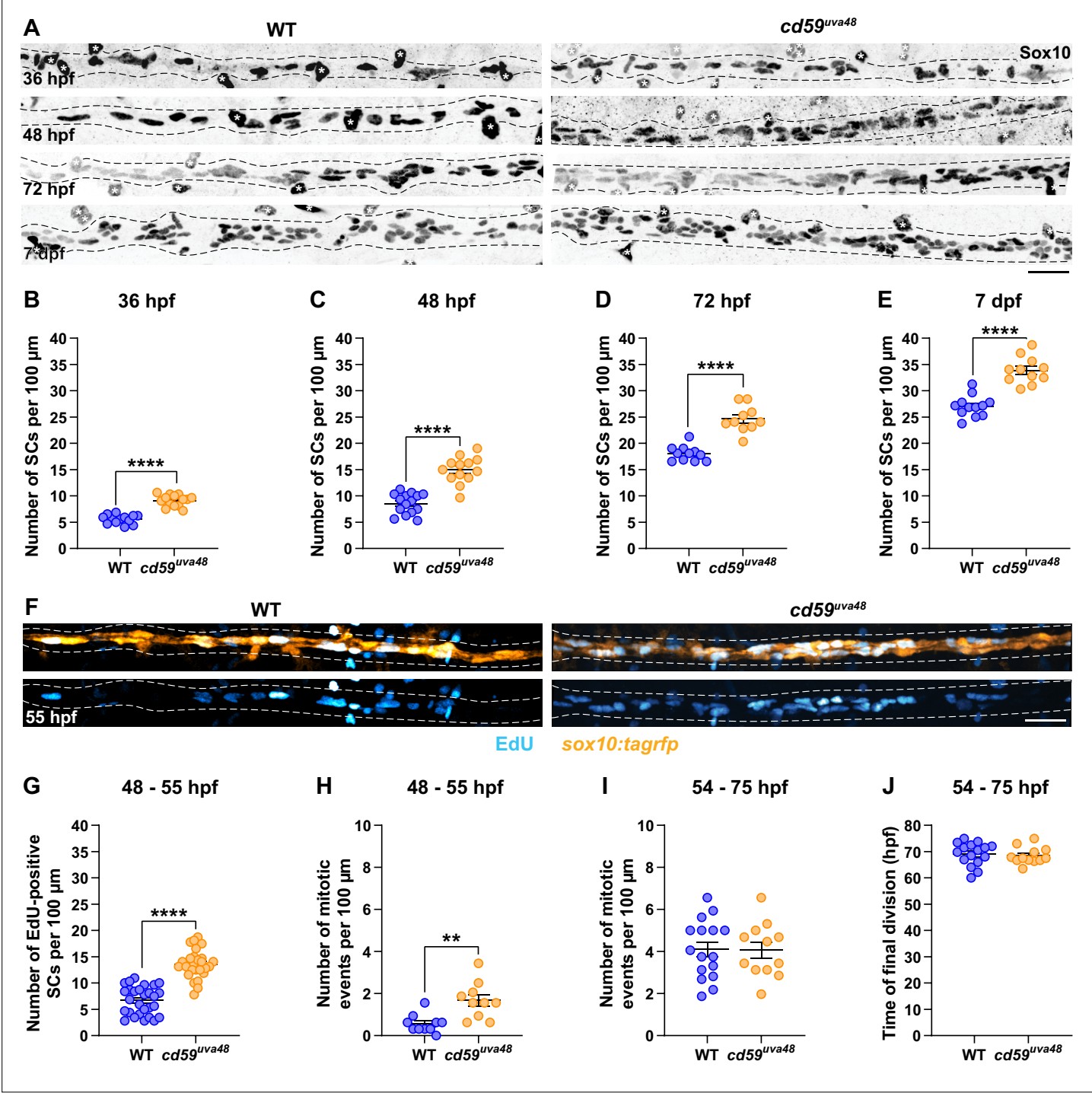

**Figure 4.** *cd59* regulates Schwann cell (SC) proliferation. (**A**) Immunofluorescence (IF) showing Sox10-positive SCs (black/gray) along the posterior lateral line nerve (pLLN) from 36 hours post fertilization (hpf) to 7 days post fertilization (dpf). Black dashed lines outline the pLLN. Sox10-positive pigment cells outside of the dashed lines (white asterisks) were not included in the analysis. (**B–E**) Scatter plots of the number of Sox10-positive SCs along the pLLN from 36 hpf to 7 dpf (mean ± SEM: 36 hpf: WT: 5.6 ± 0.2, *cd59ᵘᵛᵃ⁴⁸*: 9.1 ± 0.3; 48 hpf: WT: 8.6 ± 0.5, *cd59ᵘᵛᵃ⁴⁸*: 15.0 ± 0.7; 72 hpf: WT: 18.0 ± 0.4, *cd59ᵘᵛᵃ⁴⁸*: 24.3 ± 0.8; 7 dpf: WT: 27.0 ± 0.6, *cd59ᵘᵛᵃ⁴⁸*: 33.9 ± 0.8; p-values: p<0.0001; dot = 1 fish). (**F**) EdU incorporation assay showing *sox10:tagrfp*-positive, pLLN SCs (orange) pulsed with EdU (cyan) from 48 to 55 hpf. (**G**) Scatter plot of the number of EdU-positive SCs along the pLLN at 55 hpf (WT: 6.7 ± 0.5, *cd59ᵘᵛᵃ⁴⁸*: 13.6 ± 0.5; p<0.0001; dot = 1 fish). (**H**) Scatter plot of the number of mitotic events observed in SCs from 48 to 55 hpf (mean ± SEM: WT: 0.6 ± 0.1, *cd59ᵘᵛᵃ⁴⁸*: 1.7 ± 0.3; p=0.0019; dot = 1 fish). (**I**) Scatter plot of the number of mitotic events observed in SCs from 54 to 75 hpf (mean ± SEM: WT: 4.1 ± 0.3, *cd59ᵘᵛᵃ⁴⁸*: 4.1 ± 0.4; dot = 1 fish). (**H**) Scatter plot of the time of final cell division (hpf) observed in SCs from 54 to 75 hpf (mean ± SEM: WT:

*Figure 4 continued on next page*

*Figure 4 continued*

69.1 ± 1.1, *cd59*<sup>uva48</sup>: 68.4 ± 0.9; dot = 1 fish). All data were collected from somites 11–13 (~320 μm) and normalized to units per 100 μm. All images in this figure were acquired with confocal imaging. Each dataset was compared with Student's *t*-test using GraphPad Prism. Scale bars: (**A, F**) 25 μm.

The online version of this article includes the following source data and figure supplement(s) for figure 4:

**Source data 1.** Source data for the quantification of the number of Schwann cells (SCs) on the posterior lateral line nerve (pLLN) at 36 hours post fertilization (hpf).

**Source data 2.** Source data for the quantification of the number of Schwann cells (SCs) on the posterior lateral line nerve (pLLN) at 48 hours post fertilization (hpf).

**Source data 3.** Source data for the quantification of the number of Schwann cells (SCs) on the posterior lateral line nerve (pLLN) at 72 hours post fertilization (hpf).

**Source data 4.** Source data for the quantification of the number of Schwann cells (SCs) on the posterior lateral line nerve (pLLN) at 7 days post fertilization (dpf).

**Source data 5.** Source data for the quantification of the number of EdU-positive Schwann cells (SCs) on the posterior lateral line nerve (pLLN) from 48 to 55 hours post fertilization (hpf) depicted in *Figure 4G*.

**Source data 6.** Source data for the quantification of the number of mitotic events from 48 to 55 hours post fertilization (hpf) depicted in *Figure 4H*.

**Source data 7.** Source data for the quantification of the number of mitotic events from 54 to 75 hours post fertilization (hpf) depicted in *Figure 4I*.

**Source data 8.** Source data for the time of final Schwann cell (SC) division during 54–75 hours post fertilization (hpf) depicted in *Figure 4J*.

**Figure supplement 1.** *cd59* does not impact proliferation of neurons, neural crest cells (NCCs), motor Schwann cells (SCs), or MEP glia.

**Figure supplement 1—source data 1.** Source data for the quantification of the number of Schwann cells (SCs) on the posterior lateral line nerve (pLLN) at 48 hours post fertilization (hpf) in *cd59*<sup>uva47</sup> mutant embryos.

**Figure supplement 1—source data 2.** Source data for the quantification of the number of neurons per posterior lateral line ganglia (pLLG) at 24 hours post fertilization (hpf) depicted in *Figure 4—figure supplement 1E*.

**Figure supplement 1—source data 3.** Source data for the quantification of the number of neural crest cells (NCCs) associated with the posterior lateral line ganglia (pLLG) at 24 hours post fertilization (hpf) depicted in *Figure 4—figure supplement 1F*.

**Figure supplement 1—source data 4.** Source data for the quantification of the number of migrating neural crest cells (NCCs) per stream at 24 hours post fertilization (hpf) depicted in *Figure 4—figure supplement 1H*.

**Figure supplement 1—source data 5.** Source data for the quantification of the number of Sox10-positive cells (Schwann cells [SCs] and MEP glia) per spinal motor nerve at 72 hours post fertilization (hpf) depicted in *Figure 4—figure supplement 1J*.

**Figure supplement 1—source data 6.** Source data for the quantification of the number of trunk neuromasts at 7 days post fertilization (dpf) depicted in *Figure 4—figure supplement 1K*.

**Figure supplement 2.** Loss of *cd59* does not impact cell death within the peripheral nervous system (PNS) and central nervous system (CNS).

**Figure supplement 2—source data 1.** Source data for the quantification of the number of acridine orange (AO)-positive Schwann cells (SCs) at 48 hours post fertilization (hpf) depicted in *Figure 4—figure supplement 2B*.

**Figure supplement 2—source data 2.** Source data for the quantification of the number of acridine orange (AO)-positive cells ventral to the posterior lateral line nerve (pLLN) at 48 hours post fertilization (hpf) depicted in *Figure 4—figure supplement 2C*.

**Figure supplement 2—source data 3.** Source data for the quantification of the number of TUNEL-positive Schwann cells (SCs) at 48 hours post fertilization (hpf) depicted in *Figure 4—figure supplement 2E*.

**Figure supplement 2—source data 4.** Source data for the quantification of the number of TUNEL-positive central nervous system (CNS) cells at 48 hours post fertilization (hpf) depicted in *Figure 4—figure supplement 2F*.

observed this excess early in SC development during the SCP stage (36 hpf; *Figure 4B*) through the iSC and pmSC stages (~48 hpf, *Figure 4C*) and into mature MSC stages (72 hpf, *Figure 4D*; 7 dpf, *Figure 4E*; *Ackerman and Monk, 2016*). To rule out the possibility of off-target effects from CRISPR/Cas9 genome editing, we evaluated *cd59*<sup>uva47</sup> mutants as well. Similarly, we observed the same phenotype in mutant *cd59*<sup>uva47</sup> embryos at 48 hpf (*Figure 4—figure supplement 1A and B*), demonstrating that mutation of *cd59* was responsible for this phenotype. Together, these data demonstrate that Cd59 functions to limit overproliferation of SCs during development.

To independently confirm this excessive proliferation phenotype seen in SCs, we used EdU to label mitotically active SCs during SC development. Specifically, we incubated *sox10:tagrfp* embryos in EdU from 48 to 55 hpf. The embryos were then fixed and the number of EdU-positive SCs on the pLLN was quantified with confocal imaging. As we hypothesized, the total number of EdU-positive SCs was increased in *cd59*<sup>uva48</sup> mutant embryos compared to wildtypes (*Figure 4F and G*). From these data, we confirmed that there was more SC proliferation occurring in *cd59*<sup>uva48</sup> mutant larvae when compared

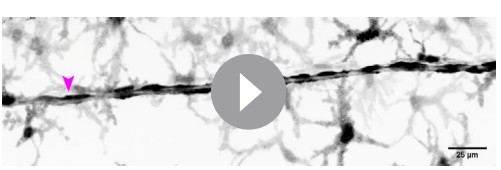

**Video 1.** Wildtype *sox10:tagrfp*-positive Schwann cells (SCs) along the posterior lateral line nerve (pLLN) migrate and undergo cell division from 48 to 55 hours post fertilization (hpf). Images were taken every 10 min, and the movie runs at 3 frames per second (fps). Data were collected from somites 11–13 (~320 μm). Mitotic events indicated with magenta arrows. Quantification of the number of mitotic events is indicated in Figure 4H. Scale bar: 25 μm.

https://elifesciences.org/articles/76640/figures#video1

**Video 3.** Wildtype *sox10:tagrfp*-positive Schwann cells (SCs) along the posterior lateral line nerve (pLLN) migrate, undergo cell division, and begin to form myelin from 54 to 75 hours post fertilization (hpf). Images were taken every 10 min, and the movie runs at 3 frames per second (fps). Data were collected from somites 11–13 (~320 μm). Mitotic events indicated with magenta arrows. Quantification of the number of mitotic events is indicated in Figure 4I. Time of last final cell division is indicated in Figure 4J. Scale bar: 25 μm.

https://elifesciences.org/articles/76640/figures#video3

to wildtype larvae. Consistent with this finding, in vivo, time-lapse imaging of *sox10:tagrfp*-positive SCs along the pLLN showed *cd59*[uva48] mutant SCs undergoing significantly more mitotic events than their wildtype siblings, with most of the increased cell divisions occurring between 48 and 55 hpf (*Figure 4H*, *Videos 1 and 2*). Between 54 and 75 hpf, however, the number of divisions was similar between wildtype and *cd59*[uva48] mutant larvae with no difference in the timing of terminal division (*Figure 4I and J*, *Videos 3 and 4*), demonstrating that, although Schwann proliferation is initially amplified in *cd59*[uva48] mutant larvae, they are capable of terminating proliferation at the same developmental stage as wildtype larvae.

Taken together, these data demonstrate that Cd59 restricts developmental SC proliferation and that excess SCs generated in *cd59* mutant larvae persist past developmental stages. To rule out changes in cell death as a contributor to SC number, we used acridine orange (AO) incorporation and TUNEL staining to quantify the number of dying SCs at 48 hpf. Co-labeling of *sox10:tagrfp*-positive SCs with AO revealed no changes in SC death between wildtype and mutant *cd59*[uva48] embryos (*Figure 4—figure supplement 2A, B*). When we assayed death more broadly in the trunk ventral to the pLLN, we also did not observe any increase in AO labeling, indicating that cell death in the embryo did not increase with *cd59* mutation (*Figure 4—figure supplement 2A and C*). Similarly, TUNEL staining combined with Sox10 labeling at 48 hpf on sectioned tissue showed that apoptotic SC death on the pLLN was unaltered in mutant *cd59*[uva48] embryos (*Figure 4—figure supplement 2D and E*). There was also no difference in the number of TUNEL-positive cells in the spinal cord (*Figure 4—figure supplement 2D and F*). Finally, in agreement with this data, we observed no SC death during in vivo imaging of *sox10:tagrfp*-positive SCs from 48 to 75 hpf in both wildtype and *cd59*[uva48] larvae (*Videos 1–4*). Overall, these findings show that *cd59* mutation does not perturb SC death.

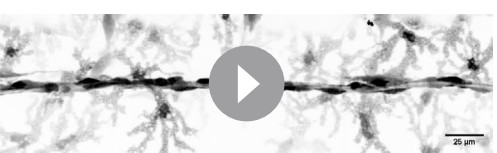

**Video 2.** *cd59*[uva48] mutant *sox10:tagrfp*-positive Schwann cells (SCs) along the posterior lateral line nerve (pLLN) migrate and undergo cell division from 48 to 55 hours post fertilization (hpf). Images were taken every 10 min, and the movie runs at 3 frames per second (fps). Data were collected from somites 11–13 (~320 μm). Mitotic events indicated with magenta arrows. Quantification of the number of mitotic events is indicated in Figure 4H. Scale bar: 25 μm.

https://elifesciences.org/articles/76640/figures#video2

**Video 4.** *cd59*[uva48] mutant *sox10:tagrfp*-positive Schwann cells (SCs) along the posterior lateral line nerve (pLLN) migrate, undergo cell division, and begin to form myelin from 54 to 75 hours post fertilization (hpf). Images were taken every 10 min, and the movie runs at 3 frames per second (fps). Data were collected from somites 11–13 (~320 μm). Mitotic events indicated with magenta arrows. Quantification of the number of mitotic events is indicated in Figure 4I. Time of last final cell division is indicated in Figure 4J. Scale bar: 25 μm.

https://elifesciences.org/articles/76640/figures#video4

Finally, we wanted to determine if *cd59* mutation affected the proliferation of other neural cells in addition to SCs. Looking at CNS neurons labeled with a HuC/HuD antibody, we observed that spinal cord neuron density was indistinguishable between wildtype and mutant *cd59*[uva48] embryos at 48 hpf (*Figure 4—figure supplement 1C*), supporting our observations that cell death does not change in the spinal cord (*Figure 4—figure supplement 2D, F*). To determine if this was also the case in the PNS, we used the same HuC/HuD antibody to label posterior lateral line ganglia (pLLG) neurons and found that the number of HuC/HuD-positive pLLG neurons was the same between wildtype and *cd58*[uva48] mutant embryos at 24 hpf (*Figure 4—figure supplement 1D, E*). These data indicate the neuronal proliferation is unaffected by *cd59* mutation. We also investigated whether NCC development was affected in our *cd59* mutant embryos. In accordance with the lack of *cd59* expression seen by RNAseq and ISH (*Figure 1—figure supplement 1A*, *Figure 2—figure supplement 1A and B*), we saw no difference in the number of cranial NCCs associated with the pLLG (*Figure 4—figure supplement 1D and F*) nor in the number of migrating spinal motor nerve NCCs (*Figure 4—figure supplement 1G and H*). Overall, these data show that Cd59 does not influence NCC proliferation. Furthermore, the increase in SC number during development was not due to an increase in the NCCs that give rise to SCs.

In addition to neurons and NCCs, we were interested in seeing whether motor nerve-associated SCs or MEP glia, which do not express *cd59* by ISH (*Figure 1—figure supplement 1A and B*), also show an overproliferation phenotype in *cd59*[uva48] mutants. To investigate this, we labeled 72 hpf *olig2:dsred* larvae with a Sox10 antibody to visualize and quantify motor nerve-associated myelinating glia (*Figure 4—figure supplement 1J*). Accuracy of this quantification was verified with quantification of *sox10:nls-eos*-positive cells along *olig2:dsred*-positive motor nerves (representative images shown in *Figure 4—figure supplement 1I*). Comparing wildtype and *cd59*[uva48] larvae, there was no significant difference between the number of Sox10-positive cells (MEP glia and SCs) along spinal motor nerves (*Figure 4—figure supplement 1I and J*). Collectively, these data show that Cd59 does not regulate proliferation of myelinating glia associated with the motor nerve.

Another important aspect of the pLLN is that it innervates neuromasts, which are sensory organs that detect water movement (*Ghysen and Dambly-Chaudière, 2007*). Trunk neuromast number and positioning changes as the zebrafish grows in length, initially starting with 5–6 primary neuromasts (deposited from 20 to 40 hpf) that eventually expand into nearly 600 neuromasts in adults (*Ghysen and Dambly-Chaudière, 2007*). Developing SCs can impact the number of neuromasts, as is evident by inappropriate neuromast formation in *erbb* pathway and *sox10* mutants which lack SCs (*Grant et al., 2005*; *Lush and Piotrowski, 2014*; *Perlin et al., 2011*; *Rojas-Muñoz et al., 2009*). To determine if excess SCs also affect neuromast formation, we labeled neuromasts with an acetylated α-Tubulin antibody and quantified the number of truncal neuromasts. Interestingly, there was no change in truncal neuromast number when comparing wildtype and *cd59*[uva48] mutant larvae at 7 dpf (*Figure 4—figure supplement 1K*), indicating that an increase in SC number does not alter neuromast formation. Combined, these data demonstrate that Cd59 is playing an early role in SC development by limiting proliferation and that this proliferative effect does not extend to other neural cells in the CNS or PNS.

## *cd59* mutant larvae may have perturbed myelin and node of Ranvier formation in the PNS

From our observations, we saw that Cd59 functions to prevent overproliferation of developing SCs. Therefore, we next wanted to understand how an increase in SC number would impact myelin and pLLN development. Proliferation is a critical aspect of radial sorting, the process by which iSCs segregate large caliber axons that require myelination, and is necessary for nerve development (*Feltri et al., 2016*). When proliferation is dysregulated, either through insufficient or excessive Schwann cell division, radial sorting can be delayed, arrested, or improperly executed, resulting in myelination that is incomplete or inappropriate, such as instances of polyaxonal myelination of small caliber axons (*Feltri et al., 2016*; *Gomez-Sanchez et al., 2009*). Therefore, we were curious how limiting SC proliferation through Cd59 influences myelinated nerve development. ISH for *mbp* (*Figure 3F*), confocal imaging of *mbpa:tagrfp-caax*-positive pLLN SCs (*Figure 5—figure supplement 1A and C*), and transmission electron microscopy (TEM) (*Figure 5A*) showed that *cd59*[uva48] mutant SCs were capable of producing myelin. Additionally, mosaic myelin labeling with *mbpa:tagrfp-caax* or *mbpa:tagrfp-caax* and *mbp:egfp-caax* constructs, injected into one-cell embryos, indicated that the myelin sheath length

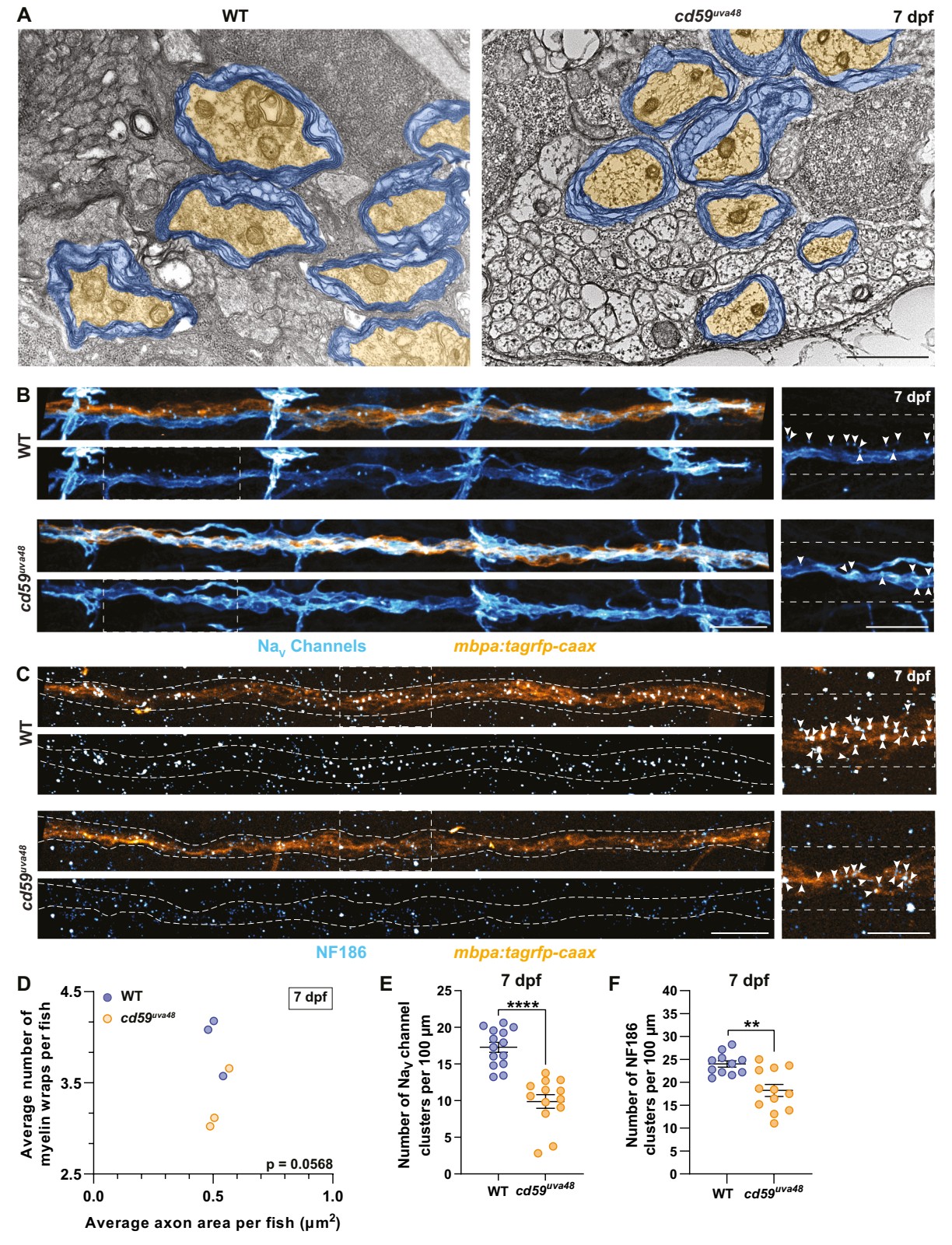

**Figure 5.** Myelin and node of Ranvier development is impaired in *cd59^uva48^* mutants. (**A**) Transmission electron micrographs showing posterior lateral line nerve (pLLN) axons myelinated by Schwann cells (SCs) at 7 days post fertilization (dpf). Myelin is shaded in blue, and myelinated axons are shaded in orange. (**B**) Immunofluorescence (IF) showing Na$_v$ channels (cyan) along *mbpa:tagrfp-caax*-positive pLLNs (orange) at 7 dpf. Diffuse Na$_v$ channel staining along unmyelinated nerves was not quantified. White dashed lines outline the pLLN. White dashed boxes correspond with the insets on the

*Figure 5 continued on next page*

*Figure 5 continued*

right. (**C**) IF showing NF186 clusters (cyan) along *mbpa:tagrfp-caax*-positive pLLNs (orange) at 7 dpf. White dashed lines outline the pLLN, and the white dashed boxes correspond to the insets on the right. Representative images in (**B**) and (**C**) depict somites 11–13 (~320 µm). (**D**) Average number of myelin wrappings per pLLN axon plotted relative to the average area of axon cross-section at 7 dpf. Data were collected from three sections per fish separated by 100 µm. Significance was determined by comparing the average number of myelin wraps divided by the average axon area for each fish with Student's *t*-test using GraphPad Prism (average number of myelin wraps per fish mean ± SEM: WT: 3.95 ± 0.19, *cd59*[uva48]: 3.27 ± 0.20; average axon area per fish mean ± SEM: WT: 0.51 ± 0.03, *cd59*[uva48]: 0.52 ± 0.03; average number of myelin wraps/average axon area per fish mean ± SEM: WT: 0.13 ± 0.01, *cd59*[uva48]: 0.16 ± 0.002; p=0.0568; dot = 1 fish). Data quantified in (**D**) were determined from electron micrographs in (**A**). (**E**) Scatter plot of the number of Na$_V$ channel clusters along *mbpa:tagrfp*-positive pLLN nerves at 7 dpf (mean ± SEM: WT: 17.3 ± 0.7, *cd59*[uva48]: 9.9 ± 0.9; p<0.0001; dot = 1 fish). (**F**) Scatter plot of the number of NF186 clusters along *mbpa:tagrfp*-positive pLLN nerves at 7 dpf (mean ± SEM: WT: 24.0 ± 0.7, *cd59*[uva48]: 18.2 ± 1.3; p=0.0011; dot = 1 fish). Data were collected from somites 3–13 (~320 µm) and normalized to units per 100 µm for (**E**) and (**F**). Images shown in (**A**) were acquired with transmission electron microscopy. Images shown in (**B**) and (**C**) were acquired with confocal imaging. Each dataset was compared with Student's *t*-test using GraphPad Prism. Scale bars: (**A**) 1 µm; (**B, C**) 25 µm.

The online version of this article includes the following source data and figure supplement(s) for figure 5:

**Source data 1.** Source data for the quantification of the average number of myelin wraps per fish relative to the average area of axon cross-section per fish at 7 days post fertilization (dpf) depicted in *Figure 5D*.

**Source data 2.** Source data for the quantification of the number of Na$_V$ channel clusters on the posterior lateral line nerve (pLLN) at 7 days post fertilization (dpf) depicted in *Figure 5E*.

**Source data 3.** Source data for the quantification of the number of NF186 clusters along the posterior lateral line nerve (pLLN) at 7 days post fertilization (dpf) depicted in *Figure 5F*.

**Figure supplement 1.** Myelin volume is reduced in *cd59*[uva48] mutants.

**Figure supplement 1—source data 1.** Source data for the quantification of myelin sheath measurements depicted in *Figure 5—figure supplement 1D*.

**Figure supplement 1—source data 2.** Source data for the quantification of myelinated nerve volume measurements depicted in *Figure 5—figure supplement 1E*.

**Figure supplement 1—source data 3.** Source data for the quantification of axon volume measurements depicted in *Figure 5—figure supplement 1F*.

**Figure supplement 1—source data 4.** Source data for the quantification of the number of myelinated axons per posterior lateral line nerve (pLLN) at 7 days post fertilization (dpf) depicted in *Figure 5—figure supplement 1G*.

at 7 dpf did not vary between wildtype and *cd59*[uva48] mutant larvae (*Figure 5—figure supplement 1A and E*) nor was there any evidence of overlapping sheaths at the same age (data not shown). However, when looking in *mbpa:tagrfp-caax* larvae, we observed that myelinated nerve volume was significantly reduced in 7 dpf *cd59*[uva48] mutant larvae (*Figure 5—figure supplement 1B and F*). Because axon volume as well as area of axon cross-sections were unaltered, as indicated by acetylated α-Tubulin labeling (*Figure 5—figure supplement 1C and G*) and TEM (*Figure 5A*, *Figure 5—figure supplement 1I*), we hypothesized that myelination was affected in *cd59*[uva48] mutants. Utilizing TEM, we compared the myelin ultrastructure in pLLNs from wildtype and *cd59*[uva48] mutant larvae at 7 dpf. From these data, we observed a nonsignificant decrease in the average number of myelin wraps around each axon relative to the average area of the axon cross-section in *cd59*[uva48] mutant larvae compared to wildtype siblings (*Figure 5A and D*). Notably, the average axon area did not change between wildtype and mutant nerves (*Figure 5A and D*) nor was there a change in the total number of myelinated axons (*Figure 5—figure supplement 1H*). Therefore, although the decrease in myelin wrapping was not statistically significant, it indicates that the decrease in myelinated nerve volume observed in *Figure 5—figure supplement 1B and F* could be related to a decrease in the number of myelin wraps on each axon. More work will be needed to determine if there is a direct relationship between myelin development and Cd59 or if this phenotype is due to Cd59-regulated SC proliferation.

In addition to producing myelin, myelinating SCs collaborate with axons to construct nodes of Ranvier, which occur between adjacent myelin sheaths and are essential for rapid neurotransmission along myelinated axons (*Rasband and Peles, 2021*). Therefore, we were curious if reduced myelin volume in *cd59*[uva48] mutant larvae would also impact nodal development. An important aspect of node construction is the clustering of axonal sodium channels, which assists in the saltatory conduction of action potentials along the nerve and is facilitated by interactions between SC-associated gliomedin and neuronal cell adhesion molecule (NrCAM) and axonal neurofascin 186 (NF186) (*Eshed et al., 2005*; *Feinberg et al., 2010*; *Rasband and Peles, 2021*; *Susuki et al., 2013*). To investigate assembly of the node, we labeled *mbpa:tagrfp-caax* larvae with antibodies to visualize sodium

channels and NF186 at 7 dpf. These studies revealed that *cd59^uva48* mutant larvae had fewer discrete sodium channel clusters along the pLLN (*Figure 5B and E*). Accordingly, mutant nerves also had fewer clusters of NF186 (*Figure 5C and F*). To confirm that our quantification was reliable, we co-labeled NF186 and sodium channels along *mbpa:tagrfp-caax*-positive nerves and observed that NF186 and sodium channels always colocalized along the myelinated nerve in wildtype larvae (*Figure 5—figure supplement 1D*), indicating that these clusters of NF186 and sodium channels are bona fide nodes of Ranvier. Considered together, these data indicate that *cd59^uva48* mutant larvae do not form nodes of Ranvier normally, though at this time it is unclear if this phenotype is a direct or indirect result of *cd59* mutation, secondary to Cd59-regulated SC proliferation, or a consequence of changes in myelin formation.

## Developmental inflammation stimulates SC proliferation and is regulated by Cd59

Cd59 is best known for its ability to inhibit complement-dependent cell lysis, protecting healthy cells from premature death during times of inflammation, such as during an infection or after an injury (*Davies et al., 1989*; *Mead et al., 2004*; *Stahel et al., 2009*; *Yao and Verkman, 2017*). Interestingly, at sublytic levels, complement can stimulate SC and OLC proliferation in vitro without inducing cell death (*Dashiell et al., 2000*; *Hila et al., 2001*; *Rus et al., 1997*; *Rus et al., 1996*; *Tatomir et al., 2020*). Complement is also a potent driver of inflammation, which is also known to drive cell proliferation (*Kiraly et al., 2015*; *Larson et al., 2020*; *Morgan, 2016*; *Morgan and Harris, 2015*; *Silva et al., 2020*). Although complement is present (*Magdalon et al., 2020*; *Zhang and Cui, 2014*), it is unclear if developmental levels of complement or inflammation could impact SC proliferation in vivo and whether this process is Cd59-dependent.

To determine if complement activity is increased in *cd59^uva48* mutant larvae, we first looked at changes in MAC-binding in developing SCs. MACs are comprised of complement proteins C5b, C6, C7, C8, and C9 and represent the culmination of the three complement pathways (classical, lectin, and alternative) (*Bayly-Jones et al., 2017*). During MAC formation, C9 will polymerize to form pores in the cell membrane, inducing cell proliferation or cell death depending on the concentration of pores (*Bayly-Jones et al., 2017*; *Morgan, 1989*; *Tegla et al., 2011b*). In healthy cells, Cd59 will bind to C8 and C9 and prevent polymerization of C9 as well as subsequent pore formation (*Meri et al., 1990*; *Ninomiya and Sims, 1992*; *Rollins et al., 1991*). Considering these mechanisms, we hypothesized that if Cd59 was dysfunctional, we would see an increase in MAC formation in SC membranes. Using an antibody against C5b8-C5b9, which recognizes assembled MACs, we observed that *cd59^uva48* mutant larvae had more MACs localized to *sox10:megfp*-positive SC membranes compared to wildtype controls at 55 hpf (*Figure 6A–C*), indicating that developing SCs are no longer protected from complement in *cd59^uva48* mutants. In the future, these results will require further investigation to confirm that this quantification reflects an increase in MAC formation after *cd59* mutation. However, these data provide preliminary evidence that there is increased MAC binding on SC membranes in *cd59^uva48* mutant embryos.

Given limited tool availability in zebrafish, we were unable to directly inhibit MAC formation in developing embryos. However, using dexamethasone (Dex), a steroid agonist that inhibits inflammation and associated complement activity (*Engelman et al., 1995*; *Silva et al., 2020*), we sought to investigate the role of developmental inflammation in Cd59-regulated SC proliferation. To do this, wildtype and mutant embryos were incubated in 1% DMSO or 1% DMSO plus 100 µM Dex from 24 to 55 hpf. This treatment method had no notable impacts on anatomical or behavioral development (*Figure 7—figure supplement 1A*). After fixing the embryos, the number of pLLN SCs was quantified using a Sox10 antibody. Excitingly, Dex treatment in *cd59^uva48* mutant embryos restored SC numbers to wildtype levels (*Figure 7A and C*). Furthermore, when compared alone, the number of SCs in wildtype larvae was also significantly reduced by Dex treatment (*Figure 7A and C*), indicating that developmental inflammation plays a role in homeostatic SC development. In support of these findings, EdU incorporation in *cd59^uva48* mutant SCs from 48 to 55 hpf also returned to wildtype levels with Dex treatment from 24 to 55 hpf (*Figure 7B and D*), confirming that Dex treatment does suppress SC proliferation. Interestingly, EdU incorporation in Dex-treated, wildtype SCs was also reduced compared to DMSO-treated, wildtype controls (*Figure 7B and D*), showing that Dex can also alter homeostatic SC proliferation. Future investigations will be needed to rule out other effects of glucocorticoid signaling

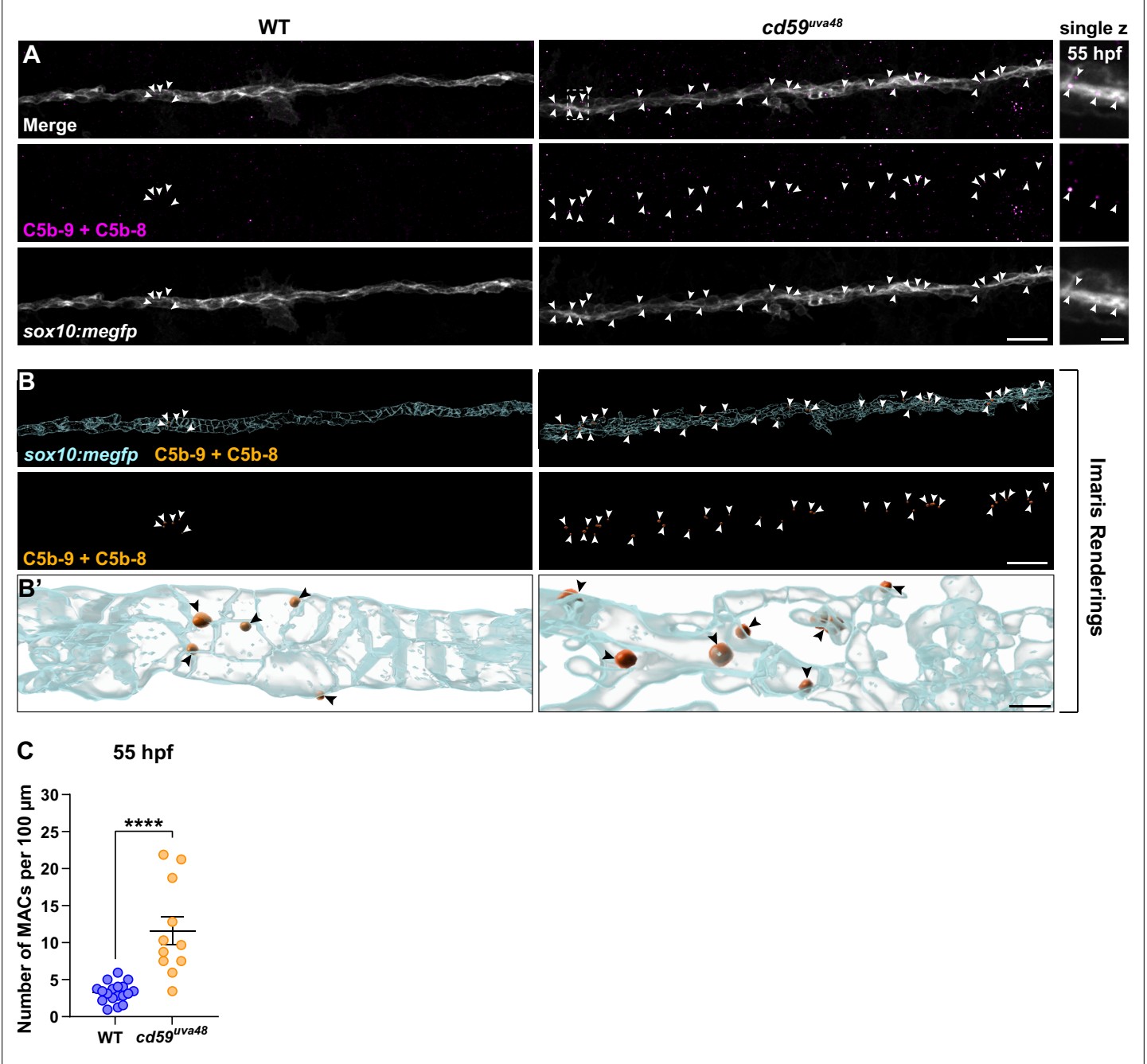

**Figure 6.** Membrane attack complex (MAC) formation on Schwann cell (SC) membranes is increased in *cd59*$^{uva48}$ mutants. (**A**) Top panel: immunofluorescence (IF) showing MACs (C5b-9+C5b-8; magenta, indicated with white arrows) embedded in *sox10:megfp*-positive posterior lateral line nerve (pLLN) SC membranes (gray) at 55 hours post fertilization (hpf). White dotted box corresponds with inset of a single z-plane on the right showing that MACs are within SC membranes. (**B**) Imaris renderings showing MACs (C5b-9+C5b-8; orange, indicated with white arrows) embedded in *sox10:megfp*-positive pLLN SC membranes (cyan) at 55 hpf. (**B′**) Enlarged renderings show MAC puncta (orange, indicated with black arrows) embedded in the SC membranes (cyan). (**C**) Scatter plot of the number of MACs in SC membranes at 55 hpf (mean ± SEM: WT: 3.3 ± 0.3, *cd59*$^{uva48}$: 11.6 ± 1.9; p<0.0001; dot = 1 fish). These data were compared with Student's *t*-test using GraphPad Prism. All data were normalized to units per 100 µm. All images were acquired with confocal imaging. Scale bars: (**A, B**) 10 µm; inset (**A**) and enlarged renderings (**B′**), 5 µm.

The online version of this article includes the following source data for figure 6:

**Source data 1.** Source data for the quantification of membrane attack complexes (MACs)-associated with Schwann cell (SC) membranes at 55 hours post fertilization (hpf) depicted in *Figure 6B*.

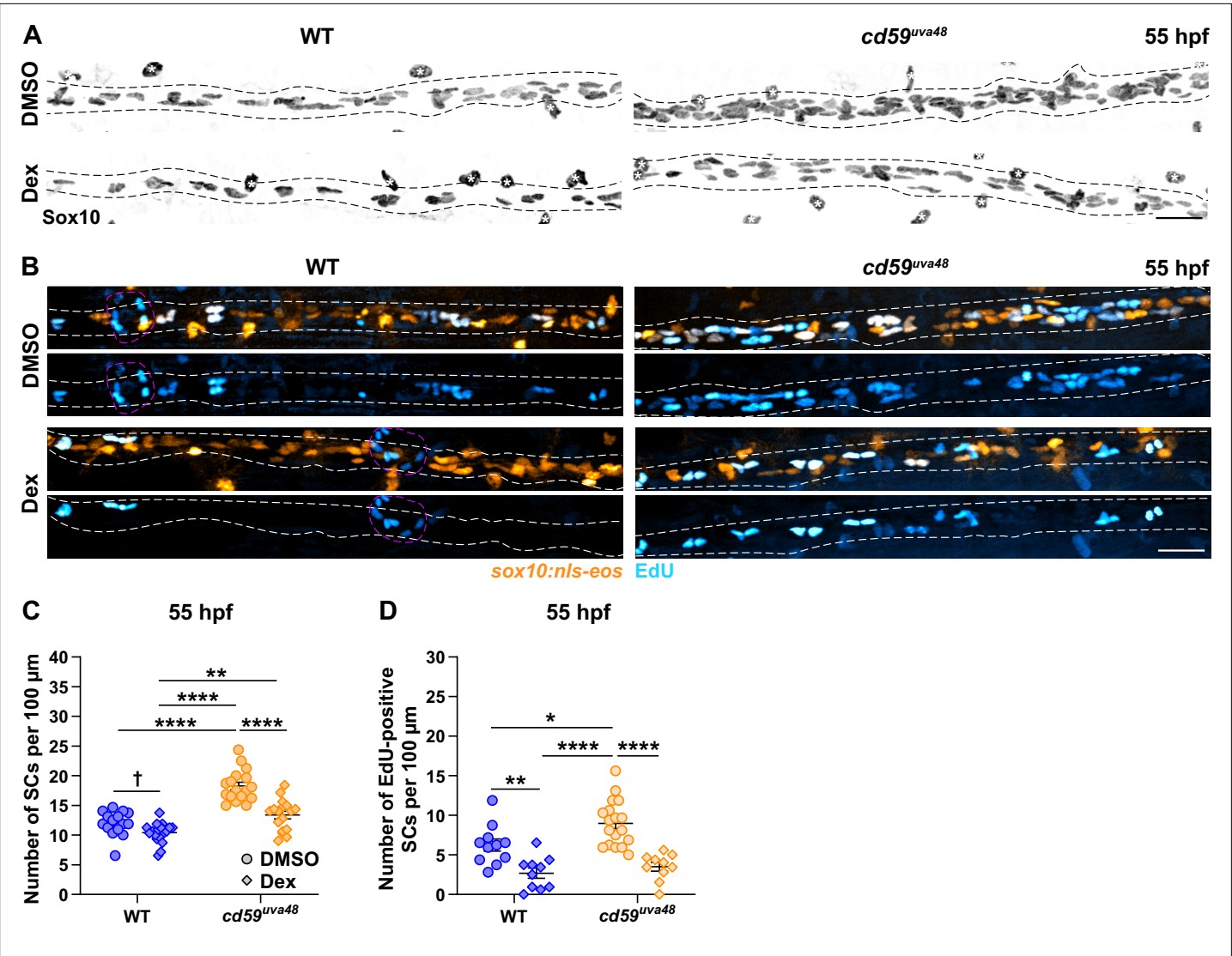

**Figure 7.** Cd59 and inflammation regulate developmental Schwann cell (SC) proliferation. (**A**) Immunofluorescence (IF) showing Sox10-positive posterior lateral line nerve (pLLN) SCs (black/gray) at 55 hours post fertilization (hpf) in embryos treated with DMSO or 100 µM dexamethasone (Dex). Black dashed lines outline the pLLN. Sox10-positive pigment cells outside of the dashed lines (white asterisks) were not included in the analysis. (**B**) EdU incorporation assay showing *sox10:nls-eos*-positive, pLLN SCs (orange) pulsed with EdU (cyan) from 48 to 55 hpf in embryos treated with DMSO or 100 µM Dex. White dashed lines outline the pLLN. Magenta dashed lines outline neuromasts. (**C**) Scatter plot of the number of pLLN SCs at 55 hpf. Asterisks (*) indicate significant differences discovered with two-way ANOVA with Tukey's post-hoc test using GraphPad Prism. Obelisk (†) indicates the significant difference discovered with Student's *t*-test (GraphPad Prism) comparing WT DMSO and WT Dex alone (mean ± SEM: DMSO: WT: 12.0 ± 0.5, *cd59*$^{uva48}$: 18.1 ± 0.6; Dex: WT: 10.4 ± 0.4, *cd59*$^{uva48}$: 13.4 ± 0.6; two-way ANOVA p-values: WT DMSO vs. *cd59*$^{uva48}$ DMSO: p<0.0001, WT Dex vs. *cd59*$^{uva48}$ DMSO: p<0.0001, WT Dex vs. *cd59*$^{uva48}$ Dex: p=0.0014, *cd59*$^{uva48}$ DMSO vs. *cd59*$^{uva48}$ Dex: p<0.0001; *t*-test p-value (compared WT only): WT DMSO vs. WT Dex: p=0.0206; dot = 1 fish). (**D**) Scatter plot of the number of EdU-positive SCs along the pLLN at 55 hpf in embryos treated with DMSO or 100 µM Dex (mean ± SEM: DMSO: WT: 4.6 ± 0.85, *cd59*$^{uva48}$: 9.0 ± 0.68; Dex: WT: 2.7 ± 0.65, *cd59*$^{uva48}$: 3.5 ± 0.53; p-values: WT DMSO vs. WT Dex: p=0.0091, WT DMSO vs. *cd59*$^{uva48}$ DMSO: p=0.0266, WT Dex vs. *cd59*$^{uva48}$ DMSO: p<0.0001, *cd59*$^{uva48}$ DMSO vs. *cd59*$^{uva48}$ Dex: p<0.0001; dot = 1 fish). These data were compared with Student's *t*-test using GraphPad Prism. All data were normalized to units per 100 µm. All images were acquired with confocal imaging. Scale bars: (**A, B**) 25 µm.

The online version of this article includes the following source data and figure supplement(s) for figure 7:

**Source data 1.** Source data for the quantification of the number of Schwann cells (SCs) on the posterior lateral line nerve (pLLN) at 55 hours post fertilization (hpf) after dexamethasone (Dex) treatment depicted in *Figure 7C*.

**Source data 2.** Source data for the quantification of of EdU-positive Schwann cells (SCs) on the posterior lateral line nerve (pLLN) at 55 hours post fertilization (hpf) after dexamethasone (Dex) treatment depicted in *Figure 7D*.

**Figure supplement 1.** Controls for dexamethasone (Dex) treatment.

in this process as well as whether or not sublytic MAC formation has a direct effect on SC proliferation in vivo; however, these data indicate that developmental inflammation aids in normal SC proliferation and that this process is amplified when *cd59* is mutated.

Previously, we showed that mutants with Cd59 dysfunction may have abnormal myelin and node of Ranvier development (*Figure 5*, *Figure 5—figure supplement 1*). Though it is still unclear if this phenotype is a direct effect of Cd59 dysfunction or secondary to overproliferation of SCs, we were curious if inhibiting inflammation-induced SC proliferation could rescue these aspects of nerve development. To investigate this hypothesis, we treated *cd59*$^{uva48}$ mutant embryos with 1% DMSO or 1% DMSO plus 100 µM Dex from 24 to 75 hpf to encompass most developmental SC proliferation. The larvae were then transferred to PTU-egg water and raised until 7 dpf. Labeling with a Sox10 antibody confirmed that this treatment regimen restored SC proliferation similarly to that described in *Figure 7A and C* (*Figure 8—figure supplement 1A and B*). We then repeated the same treatment procedure in wildtype and *cd59*$^{uva48}$ mutant *mbpa:tagrfp-caax* embryos and quantified sodium channel antibody labeling at 7 dpf. Remarkably, we observed that Dex treatment dramatically increased the number of sodium channel clusters in *cd59*$^{uva48}$ mutant larvae, achieving cluster levels similar to wildtype siblings (*Figure 8A and C*). Similarly, when comparing myelinated nerve volume in *mbpa:tagrfp-caax* larvae at 7 dpf, we saw that Dex could also restore *cd59*$^{uva48}$ mutant nerve volume to wildtype levels (*Figure 8B and D*). Collectively, these data indicate that inflammation-induced SC proliferation contributes to perturbed myelin and node of Ranvier development. Although more work will be needed to determine where Cd59 and MACs fit into this process, these experiments provide preliminary evidence that inhibition of developmental inflammation can protect nerve development after *cd59* mutation.

## Discussion

Myelination during nervous system development is essential for neural function, providing trophic and structural support to axons as well as quickening electrical conduction (*Ritchie, 1982*; *Stadelmann et al., 2019*). Consequently, impairment of this process can be devastating to patient quality of life (*Stadelmann et al., 2019*; *van der Knaap and Bugiani, 2017*). For this reason, the genetic mechanisms that orchestrate myelinating glial cell development requires continued exploration. Over the past few decades, we and others have noted expression of *cd59* that is conserved in developing myelinating glial cells across multiple organisms, including zebrafish, rodents, and humans (*Gerber et al., 2021*; *Howard et al., 2021*; *Marisca et al., 2020*; *Marques et al., 2018*; *Marques et al., 2016*; *Piller et al., 2021*; *Saunders et al., 2019*; *Siems et al., 2021*; *Sun et al., 2013*; *Wolbert et al., 2020*; *Zajicek et al., 1995*; *Zhu et al., 2019*). Despite these observations, there had been little exploration into the function of Cd59 in the developing nervous system. In this study, we demonstrate that *cd59* is expressed in a subset of developing SCs as well as mature OLs and SCs, revealing transcriptional heterogeneity among myelinating glial cells during development. Focusing on SCs, we demonstrated that Cd59 regulates SC proliferation induced by developmental inflammation. Furthermore, embryos with Cd59 dysfunction may have abnormal myelin and node of Ranvier formation during development. Overall, these findings illuminate the intersection of the innate immune system and glial cells and how they collaborate to establish a functioning nervous system during development.

Considering the importance of myelination, our investigation revealed some interesting myelin defects in response to *cd59* dysfunction. With the increase in SC proliferation in *cd59* mutant embryos, we hypothesized that the myelin sheath lengths would shorten in order to accommodate the excess SCs and maintain the same amount of myelin on the nerve. Unexpectedly, we noted that with Cd59 dysfunction, the myelin sheath length was unaffected. Additionally, myelin volume was reduced, which may be related to a nonsignificant decrease in the average number myelin wraps around each axon. At this point, it is unclear if the decrease in myelin volume is directly due to Cd59 dysfunction, secondary to SC excess, or simply an indirect consequence of developmental inflammation. That said, from these data we can imagine a couple of potential hypotheses as to why myelin volume is decreased. First, it is possible that Cd59, in addition to regulating SC proliferation, is also necessary for myelinogenesis. Past investigations demonstrate that CD59a helps facilitate vesicle signaling, which is important during insulin release and is implicated in diabetic patients (*Golec et al., 2019*; *Krus et al., 2014*). Within the nervous system, vesicle-dependent signaling is also important for myelin formation as well as trafficking within the myelin sheath (*Baron and Hoekstra, 2010*; *Reiter and Bongarzone, 2020*; *White and Krämer-Albers, 2014*). Electron micrographs show that CD59a is positioned throughout

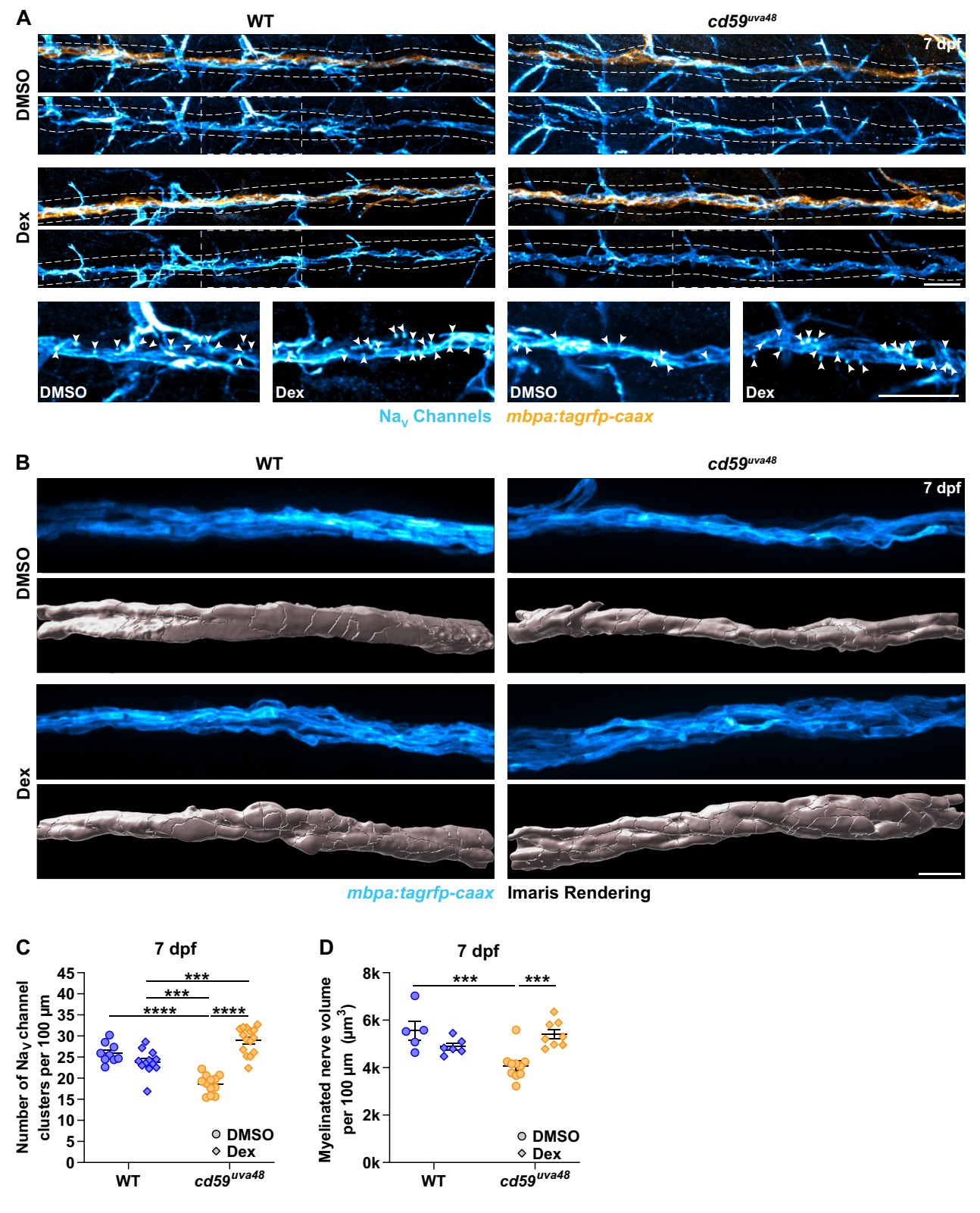

Na_v Channels   *mbpa:tagrfp-caax*

*mbpa:tagrfp-caax*   Imaris Rendering

**Figure 8.** Developmental inflammation influences myelin and node of Ranvier development. (**A**) Immunofluorescence (IF) showing Na_v channels (cyan) along *mbpa:tagrfp-caax*-positive nerves (orange) at 7 days post fertilization (dpf) in larvae treated with DMSO or 100 µM dexamethasone (Dex). Diffuse Na_v channel staining along unmyelinated nerves was not quantified. White dashed lines outline the posterior lateral line nerve (pLLN). White dashed boxes correspond with the insets below. Representative images are from somite 11–13 (~320 µm). (**B**) In vivo imaging showing the volume of

*Figure 8 continued on next page*

*Figure 8 continued*

*mbpa:tagrfp-caax*-positive nerves at 7 dpf in larvae treated with DMSO or 100 µM Dex. Bottom panels depict Imaris renderings (white) of myelinated nerve volumes. Representative images are from somite 12 (~110 µm). (**C**) Scatter plot of the number of Na$_V$ channel clusters along *mbpa:tagrfp-caax*-positive nerves at 7 dpf (mean ± SEM: DMSO: WT: 26.0 ± 0.8, *cd59*$^{uva48}$: 18.6 ± 0.6; Dex: WT: 23.9 ± 0.8, *cd59*$^{uva48}$: 28.9 ± 0.8; p-values: WT DMSO vs. *cd59*$^{uva48}$ DMSO: p<0.0001, WT Dex vs. *cd59*$^{uva48}$ DMSO: p=0.0001, WT Dex vs. *cd59*$^{uva48}$ Dex: p=0.0001, *cd59*$^{uva48}$ DMSO vs. *cd59*$^{uva48}$ Dex: p<0.0001; dot = 1 fish). These data were compared with a two-way ANOVA with Tukey's post-hoc test using GraphPad Prism. Data were collected from somites 3–13 (~320 µm). (**D**) Scatter plot of myelinated nerve volumes at 7 dpf (mean ± SEM: DMSO: WT: 5.6 ± 0.4, *cd59*$^{uva48}$: 4.1 ± 0.2; Dex: WT: 4.9 ± 0.1, *cd59*$^{uva48}$: 5.4 ± 0.2; p--values: WT DMSO vs. *cd59*$^{uva48}$ DMSO: p=0.0009, *cd59*$^{uva48}$ DMSO vs. *cd59*$^{uva48}$ Dex: p=0.0006; dot = 1 fish). These data were compared with a two-way ANOVA with Tukey's post-hoc test using GraphPad Prism. Data were collected from somite 12 (~110 µm). All data were normalized to units per 100 µm. All images were acquired with confocal imaging. Scale bars: (**A**) 25 µm; inset, 25 µm; (**B**) 10 µm.

The online version of this article includes the following source data and figure supplement(s) for figure 8:

**Source data 1.** Source data for the quantification of the number of Na$_V$ channel clusters with dexamethasone (Dex) treatment depicted in *Figure 8C*.

**Source data 2.** Source data for the quantification of the myelinated nerve volume with dexamethasone (Dex) treatment depicted in *Figure 8D*.

**Figure supplement 1.** Extended dexamethasone (Dex) treatment had same impact on Schwann cell (SC) proliferation in *cd59*$^{uva48}$ mutant larvae.

**Figure supplement 1—source data 1.** Source data for the quantification of the number of Schwann cells (SCs) on the posterior lateral line nerve (pLLN) at 7 days post fertilization (dpf) after extended dexamethasone (Dex) treatment depicted in *Figure 8—figure supplement 1B*.

the myelin sheath in oligodendrocytes (*Siems et al., 2021*). Notably, CD59a molecules are observed deep within compacted myelin (*Siems et al., 2021*). Because Cd59 is unlikely to encounter complement when it is so isolated from the extracellular environment, it is possible that Cd59 is playing an additional role in mature myelinating glial cells, possibly by regulating vesicle signaling within the myelin sheath. In future work, it will be interesting to identify if Cd59 has any direct role in SC and/ or oligodendrocyte myelinogenesis. That said, it is important to note that *cd59* mutant SCs continue to express molecules necessary for SC development and subsequent myelination, including *sox10*, *mbp*, and *krox20* (*Bremer et al., 2011*; *Gould et al., 1995*; *Jessen and Mirsky, 2005*; *Takada et al., 2010*; *Topilko et al., 1994*), as well as produce myelin at the appropriate developmental stage. Therefore, these SCs are able to make myelin without Cd59. These facts considered, it is also possible that Cd59 is not directly involved in SC myelinogenesis. With this in mind, another possibility is that the excess SCs themselves are responsible for the reduction in myelin through a contact inhibition mechanism. In support of this idea, prior work in vitro demonstrates that SCs produce a glycoprotein called contactinhibin when the cells reach a high density (*Casella et al., 2000*). Contactinhibin and its receptor, contactinhibin receptor, mediate contact-dependent inhibition of growth in SCs and fibroblasts (*Casella et al., 2000*; *Gradl et al., 1995*; *Wieser et al., 1990*), demonstrating that SCs have methods to restrict overgrowth under homeostatic conditions. Interestingly, contactinhibin also interacts with vimentin (*Wieser et al., 1990*), which is known to collaborate with TACE to negatively regulate NRG1 type III expression and subsequent myelin production (*Triolo et al., 2012*). Without vimentin and TACE, SCs hypermyelinate nerves (*Triolo et al., 2012*). Considering these findings, it is possible that overproliferation of SCs in *cd59* mutant larvae leads to activation of these growth-inhibiting molecules, such as contactinhibin, vimentin, and TACE, and consequently decreases NRG1 type III expression and myelinogenesis. Finally, as mentioned previously, it is also possible that Cd59 or Cd59-regulated SC proliferation has nothing to do with changes in myelin development. Dex treatment indicates that developmental inflammation (or possibly other effects of glucocorticoid signaling) contributes to abnormal myelin formation in *cd59* mutant larvae. Therefore, it is possible that this altered myelination is an indirect effect of developmental inflammation and not relevant to Cd59 or Cd59-regulated SC proliferation, indicating that we have more to learn about the role of developmental inflammation in myelinogenesis. Utilizing the *cd59* mutant model established in this article, future studies can investigate these hypotheses to learn more about the molecular mechanisms that regulate myelin development.

While evaluating myelination, we were also curious about the SCs that were not contributing to myelination. Because the number of myelinated axons as well as myelin sheath length is similar between wildtype and mutant nerves, this indicates that the extra SCs are not producing myelin. It is unclear if this extra population of SCs are NMSCs or undifferentiated iSCs, which is a question worth exploring in a future study. Fortunately, a recent transcriptomic analysis of developing SCs revealed several new markers in which to better differentiate iSCs, NMSCs, and MSCs (*Tasdemir-Yilmaz et al.,*

*2021*), which have historically been difficult to distinguish. Using these new cell-specific markers, future investigations will be able to determine the fate of these extra SCs on *cd59* mutant nerves.

Beyond changes in myelination, in this study we showed that Cd59-limited proliferation is elicited by developmental inflammation. This finding provokes many interesting questions. First, these data reiterate the idea that the innate immune system and genes traditionally active in immune cells are first used during development to guide nervous system assembly and formation. In the CNS, there is evidence that complement aids in stimulating synaptic pruning of developing dendrites, directing cell polarity in the ventricular zone, guiding cortical neuron migration, and fostering neural progenitor cell proliferation (*Coulthard et al., 2018*; *Coulthard et al., 2017*; *Denny et al., 2013*; *Gorelik et al., 2017*; *Magdalon et al., 2020*; *Paolicelli et al., 2011*; *Stevens et al., 2007*). Similarly, inflammasome signaling was recently shown to be a necessary asset in Purkinje neuron development and mutations in this pathway are associated with increased DNA damage and behavioral deficits (*Lammert et al., 2020*). Finally, microglia, the resident innate immune cell of the CNS, have several roles in CNS development, including phagocytosing cell debris as well as pruning developing synapses and myelin (*Hughes and Appel, 2020*; *Mazaheri et al., 2014*; *Silva et al., 2021*; *Stevens et al., 2007*; *Villani et al., 2019*). To our knowledge, our study is the first examination of the role of inflammation and complement signaling in PNS development beyond the NCC stage (*Carmona-Fontaine et al., 2011*). These data prompt future exploration into this relationship between the nervous and immune systems during PNS formation.

To further this point, our data indicate that MACs may be involved in enhancing SC proliferation during development. Previous findings in vitro demonstrated that sublytic MACs could stimulate SC proliferation through activation of the MAPK/ERK and PKC pathways (*Dashiell et al., 2000*; *Hila et al., 2001*). These and other mitogenic pathways are also activated in other cells that undergo MAC-stimulated proliferation (*Tegla et al., 2011b*), such as smooth muscle cells, T cells, and oligodendrocytes (*Chauhan and Moore, 2011*; *Rus et al., 1997*; *Rus et al., 1996*; *Tegla et al., 2011b*; *Zwaka et al., 2003*). Collectively, these data have established that MACs induce cell proliferation through activation of Gi protein and Ras/Raf1, which go onto activate MEK1 and ERK1 that then facilitate transcription of cell cycle-related genes, such as CDK4 and CDK2 (*Dashiell et al., 2000*; *Niculescu et al., 1999*; *Niculescu et al., 1997*; *Tegla et al., 2011b*). Simultaneously, PI3K and Akt activate mTOR, $P70^{S6}$, and $K_V1.3$ to induce DNA and protein synthesis that aids in cell cycle activation (*Badea et al., 2002*; *Hila et al., 2001*; *Rus et al., 2001*; *Tegla et al., 2011a*; *Tegla et al., 2011b*). At this point, it is unclear whether increased MAC formation in *cd59* mutant SCs leads to activation of these pathways in vivo and whether this mechanism contributes to developmental SC proliferation. Utilizing the tools developed in this study, we look forward to future investigations of these pathways and their role in SC development.

During our investigation, we also show that *cd59* is expressed in a subset of SCs and OLs and is not expressed in other myelinating glial cells, including MEP glia and SCs that associate with spinal motor nerves. This expression pattern persists at least until 7 dpf in SCs, indicating that developmental heterogeneity lingers in the mature SCs. Furthermore, we show that Cd59 regulates proliferation of sensory SCs on the pLLN but not motor nerve-associated SCs, indicating that these differences in *cd59* expression are associated with functional heterogeneity as well. These findings provoke many topics for further investigation. First, how do *cd59*-positive SCs differ from *cd59*-negative SCs? Does this imply that a subset of SCs is more sensitive to complement activity, or are there other implications for these expression differences? Related, why do motor SCs lack *cd59*? Recent RNAseq analysis of developing satellite glial cells (SGCs) and SCs showed that glial cell precursors, SGCs, and iSCs are heterogenous and that this transcriptional diversity depended on the type of nerve/ganglia they associated with (*Tasdemir-Yilmaz et al., 2021*; *Wiltbank and Kucenas, 2021*). Motor neurons and sensory neurons are transcriptionally and functionally distinct and likely have different demands of the SCs that they are intimately associated with. Therefore, it follows that sensory SC functionality may require Cd59 whereas motor SCs do not. It will be interesting to explore the consequences of this heterogeneity in future investigations.

We are also curious if Cd59 function is multifaceted in the nervous system. Our findings demonstrated that Cd59 prevents overproliferation of SCs during development by shielding them from developmental inflammation. Similar observations have been noted in T cells and smooth muscle cells (*Li et al., 2013*; *Longhi et al., 2005*). That said, many other functions beyond proliferation

control have been documented for CD59. For example, CD59 is required to instruct proximal-distal cell identity, a process that is necessary for proper cell positioning during limb regeneration (*Echeverri and Tanaka, 2005*). In our study, we noted that floor plate and hypochord cells express *cd59*. During neural tube development, floor plate cells play an important role in determining cell fate as well as dorsal-ventral patterning in the spinal cord (*Hirano et al., 1991*; *Yu et al., 2013*), whereas the hypochord orchestrates midline blood vessel pattering (*Cleaver and Krieg, 1998*). Considering the role of CD59 in dictating proximal-distal cell identity during limb regeneration (*Echeverri and Tanaka, 2005*), it would be interesting to see if Cd59 participated in similar signaling pathways in floor plate and hypochord cells. Alternatively, Cd59's role in dictating proximal-distal cell identity could also be useful during nerve regeneration. Like limb regeneration, there needs to be an appropriate distribution of cells along the proximal-distal axis of the regenerating nerve. With this idea in mind, it would be interesting to examine how Cd59 dysfunction impacts SC distribution after nerve regeneration.

Finally, our study revealed that Cd59 plays a role in nervous system development. Previous case studies of patients with CD59 dysfunction, such as those with germline PNH and congenital CD59 deficiency, noted early-onset neurological dysfunction in these patients (*Haliloglu et al., 2015*; *Höchsmann and Schrezenmeier, 2015*; *Johnston et al., 2012*; *Karbian et al., 2018*; *Solmaz et al., 2020*). These neurological symptoms were generally attributed to damage from overactivation of complement and excess inflammation (*Höchsmann and Schrezenmeier, 2015*). In support of this idea, eculizumab treatment, which inhibits complement protein 5 (C5) and limits formation of MACs, slowed the progression of severe neurological symptoms, including bulbar symptoms, focal seizures, impaired respiration, and muscular hypotonia. However, eculizumab could not restore full neurological function in these patients (*Höchsmann and Schrezenmeier, 2015*). In light of our recent findings, we propose that CD59 dysfunction could be interfering with nervous system development in these human patients, likely through inflammation-induced overproliferation of SCs and subsequent malformation of myelin and nodes of Ranvier. However, because developmental SC proliferation has completed by the time these patients receive eculizumab (*Cravioto, 1965*; *Höchsmann and Schrezenmeier, 2015*), it is likely too late to restore SC proliferation and preserve nerve development. In future case studies of these patients, it would be interesting to confirm with nerve biopsies whether these patients also have excess SCs and abnormal myelin volume like we have observed in our zebrafish model. In support of this idea, neutrophils isolated from patients with PNH also overproliferate (*Li et al., 2019*), indicating that SCs may do the same. Ultimately, there remain many areas of exploration to fully characterize the role of Cd59 in the developing and mature nervous system.

# Materials and methods

## Key resources table

| Reagent type (species) or resource | Designation | Source or reference | Identifiers | Additional information |
|---|---|---|---|---|
| Strain, strain background (*Danio rerio*) | AB* | ZIRC | RRID:ZFIN_ZDB-GENO-960809-7 | |
| Genetic reagent (*D. rerio*) | Tg(sox10(4.9):nls-eos)[w18] | *McGraw et al., 2012* | RRID:ZFIN_ZDB-ALT-110721-2 | |
| Genetic reagent (*D. rerio*) | Tg(sox10(4.9):tagrfp)[uva5] | *Zhu et al., 2019* | RRID:ZFIN_ZDB-ALT-200513-7 | Also referred to as *sox10:tagrfp*; cytoplasmic expression of TagRFP |
| Genetic reagent (*D. rerio*) | Tg(sox10(7.2):megfp)[sl3] | *Kirby et al., 2006* | RRID:ZFIN_ZDB-ALT-150113-6 | Also referred to as *sox10:megfp*; membrane-tethered expression of eGFP |
| Genetic reagent (*D. rerio*) | Tg(mbp(2.0):egfp-caax)[ue2] | *Almeida et al., 2011* | RRID:ZFIN_ZDB-ALT-120103-2 | Also referred to as *mbp:egfp-caax*; membrane-tethered expression of eGFP |
| Genetic reagent (*D. rerio*) | gSAIzGFFD37A | *Brown et al., 2022* | | Also referred to as *Gt(erbb3b:gal4);Tg(uas:egfp)* or *gal4:erbb3b;uas:egfp*; cytoplasmic expression of eGFP |
| Genetic reagent (*D. rerio*) | Tg(olig2:dsred2)[vu19] | *Shin et al., 2003* | RRID:ZFIN_ZDB-FISH-150901-8168 | Also referred to as *olig2:dsred*; cytoplasmic expression of DsRed2 |
| Genetic reagent (*D. rerio*) | Tg(mbpa(6.6):tagrfp-caax;cry:egfp)[uva53] | This paper | | Also referred to as *mbpa:tagrfp-caax*; membrane expression of TagRFP; more information found in 'Generation of transgenic lines'; available from the Kucenas Lab |
| Genetic reagent (*D. rerio*) | Tg(cd59(5.0):tagrfp)[uva52] | This paper | | Also referred to as *cd59:tagrfp*; cytoplasmic expression of TagRFP; more information found in 'Generation of transgenic lines'; available from the Kucenas Lab |

*Continued on next page*

*Continued*

| Reagent type (species) or resource | Designation | Source or reference | Identifiers | Additional information |
|---|---|---|---|---|
| Genetic reagent (*D. rerio*) | *cd59*[uva48] | This paper | | 15 bp deletion at splice site between exon 2 and intron 2 of *cd59* gene, BX957297.10:g.44_58delTGCTGGGGCTTGGTA; more information found in 'Generation of mutant lines'; available from the Kucenas Lab |
| Genetic reagent (*D. rerio*) | *cd59*[uva47] | This paper | | 6 bp deletion in exon 2 of *cd59* gene, BX957297.10:g.45_50delGCTGGG; more information found in 'Generation of mutant lines'; available from the Kucenas Lab |
| Recombinant DNA reagent | p5E-cd59(−5.0) | This paper | | More information found in 'Generation of transgenic lines'; available from the Kucenas Lab |
| Recombinant DNA reagent | p5E-mbpa(−6.6) | This paper | | More information found in 'Generation of transgenic lines'; available from the Kucenas Lab |
| Recombinant DNA reagent | pME-tagrfp | *Don et al., 2017* | N/A | |
| Recombinant DNA reagent | pME-tagrfpcaax | *Auer et al., 2015* | N/A | |
| Recombinant DNA reagent | p3E-polyA | *Kwan et al., 2007* | N/A | |
| Recombinant DNA reagent | pDestTol2pA2 | *Kwan et al., 2007* | N/A | |
| Recombinant DNA reagent | pDestTol2pA2cryegfp | *Kwan et al., 2007* | N/A | |
| Recombinant DNA reagent | pCS2FA-transposase | *Kwan et al., 2007* | N/A | Template for *Tol2* transposase mRNA synthesis |
| Commercial assay or kit | MEGAshortscript T7 transcription kit | Invitrogen | Cat# AM1354 | |
| Commercial assay or kit | QIAprep spin miniprep kit | QIAGEN | Cat# 27106 | |
| Commercial assay or kit | QIAquick PCR purification kit | QIAGEN | Cat# 28106 | |
| Commercial assay or kit | QIAquick gel extraction kit | QIAGEN | Cat# 28704 | |
| Commercial assay or kit | RNeasy mini kit | QIAGEN | Cat# 74104 | |
| Commercial assay or kit | pENTR 5′-TOPO cloning kit | Invitrogen | Cat# K59120 | |
| Commercial assay or kit | TOPO TA cloning kit | Invitrogen | Cat# K4575J10 | |
| Commercial assay or kit | LR clonase II plus | Invitrogen | Cat# 12538-120 | |
| Commercial assay or kit | Click-it EdU cell proliferation kit for imaging. Alexa Fluor 647 dye | Invitrogen | Cat# C11340 | |
| Commercial assay or kit | RNAscope fluorescent multiplex reagent kit | ACD | Cat# 320850 | |
| Commercial assay or kit | mMESSAGE mMACHINE sp6 transcription kit | Invitrogen | Cat# AM1340 | |
| Commercial assay or kit | ApopTag red in situ apoptosis detection kit | Sigma | Cat# S7165 | |
| Commercial assay or kit | High-capacity cDNA reverse transcription kit | Thermo Fisher | Cat# 4368814 | |
| Commercial assay or kit | Glutaraldehyde (electron microscopy grade) | Sigma | Cat# G7651 | |
| Chemical compound, drug | RNAscope probe diluent | ACD | Cat# 300041 | |
| Chemical compound, drug | RNAscope probe-Dr-cd59-C2 | ACD | Cat# 561561-C2 | |
| Chemical compound, drug | DAPI fluoromount-G | Southern Biotech | Cat# AM1340 | |

*Continued on next page*

*Continued*

| Reagent type (species) or resource | Designation | Source or reference | Identifiers | Additional information |
|---|---|---|---|---|
| Chemical compound, drug | DIG RNA labeling mix | Roche | Cat# 11277073910 | |
| Chemical compound, drug | Dexamethasone | Sigma | Cat# D1756 | 100 µM, also referred to as Dex |
| Chemical compound, drug | Instant ocean sea salt | ThatFishPlace | Cat# 242818 | 0.3 g/L, used to make egg water |
| Chemical compound, drug | 1-Phenyl-2-thiourea | Sigma | Cat# P7629 | 0.004%, also referred to as PTU |
| Chemical compound, drug | Tricaine-S (MS-222) | The Pond Outlet | No Cat# | Also referred to as tricaine |
| Chemical compound, drug | Cas9 protein | PNA Bio | Cat# CP01-50 | |
| Chemical compound, drug | Low gelling temperature agarose | Sigma | Cat# 9414 | |
| Chemical compound, drug | Agar | Fisher Scientific | Cat# BP1423500 | |
| Chemical compound, drug | Sucrose | Sigma | Cat# S5016 | |
| Chemical compound, drug | 2-Methylbutane | Fisher Scientific | Cat# 03551-4 | |
| Chemical compound, drug | Acridine orange hemi (zinc chloride) salt | Santa Cruz Biotechnology | Cat# sc-214488 | Also referred to as AO |
| Chemical compound, drug | Diethyl pyrocarbonate | Sigma | Cat# D5758 | 1:1000; also referred to as DEPC |
| Chemical compound, drug | 20× SSC | Quality Biological | Cat# 351-003-131 | |
| Chemical compound, drug | Triton X-100 | Sigma | Cat# T8787 | |
| Chemical compound, drug | 100 bp DNA ladder | New England BioLabs Inc. | Cat# N3231L | |
| Chemical compound, drug | Sodium cacodylate trihydrate | Sigma | Cat# 0250 | Also referred to as SCT |
| Chemical compound, drug | Osmium tetroxide (electron microscopy grade) | Sigma | Cat# 75632 | |
| Chemical compound, drug | Uranyl acetate | *Morris et al., 2017* | | |
| Chemical compound, drug | EPON | *Morris et al., 2017* | | |
| Chemical compound, drug | Sheep serum | Gemini Bioproducts | Cat# 100-117 | |
| Chemical compound, drug | Goat serum | Gemini Bioproducts | Cat# 100-109 | |
| Chemical compound, drug | Proteinase K | Fisher Scientific | Cat# BP1700-100 | |

*Continued on next page*

*Continued*

| Reagent type (species) or resource | Designation | Source or reference | Identifiers | Additional information |
|---|---|---|---|---|
| Chemical compound, drug | Bovine serum albumin | Fisher Scientific | Cat# BP1600-100 | |
| Antibody | Anti-HuC/HuD (mouse monoclonal) | Invitrogen | Cat# A-21271 | (1:500) |
| Antibody | Anti-acetylated tubulin (mouse monoclonal) | Sigma | Cat# T7451 | (1:10,000); acetylated tubulin; also referred to as tubulin |
| Antibody | Fab fragments anti-digoxigenin-AP (sheep polyclonal) | Sigma | Cat# 11093274910; RRID:AB_514497 | (1:5000) |
| Antibody | Anti-sox10 (rabbit polyclonal) | *Binari et al., 2013* | N/A | (1:5000) |
| Antibody | Anti-GFP (chicken polyclonal) | Abcam | Cat# ab13970; RRID:AB_300798 | (1:500) |
| Antibody | Anti-sodium channel, pan (mouse monoclonal) | Sigma | Cat# S8809 | (1:500) |
| Antibody | Anti-TagRFP (rabbit polyclonal) | Invitrogen | Cat# 10367 | (1:500) |
| Antibody | Anti-neurofascin 186 (rabbit polyclonal) | Gift from Matthew Rasband | N/A | (1:200), neurofascin 186; also referred to as NF186 |
| Antibody | Anti-C5b-8+C5b-9 (mouse monoclonal) | Abcam | Cat# ab66768 | (1:500), C5b-8+C5b-9; also referred to membrane attack complex or MAC |
| Antibody | Alexa Fluor 488 anti-chicken IgY (H+L) (goat polyclonal) | Thermo Fisher | Cat# A-11039; RRID:AB_2534096 | (1:1000) |
| Antibody | Alexa Fluor 647 anti-rabbit IgG (H+L) (goat polyclonal) | Thermo Fisher | Cat# A-21244; RRID:AB_2535812 | (1:1000) |
| Antibody | Alexa Fluor 647 anti-mouse IgG (H+L) (goat polyclonal) | Thermo Fisher | Cat# A-21235; RRID:AB_2535804 | (1:1000) |
| Antibody | Alexa Fluor 488 anti-mouse IgG (H+L) (goat polyclonal) | Thermo Fisher | Cat# A-11001 | (1:1000) |
| Antibody | Alexa Fluor 568 anti-rabbit IgG (H+L) (goat polyclonal) | Thermo Fisher | Cat# A-11011 | (1:1000) |
| Sequence-based reagent | cd59-F (for transgenic construction) | This paper | PCR primers | 5'-TCAGATCACATCACACCTGA-3'; more information found in 'Generation of transgenic lines' |
| Sequence-based reagent | cd59-R (for transgenic construction) | This paper | PCR primers | 5'-AATGCCTTCAGTTTACCAGTCT-3'; more information found in 'Generation of transgenic lines' |
| Sequence-based reagent | mbpa-F (for transgenic construction) | This paper | PCR primers | 5'-ATGTCGAGTAATATCGAGCAGC-3'; more information found in 'Generation of transgenic lines' |
| Sequence-based reagent | mbpa-R (for transgenic construction) | *Almeida et al., 2011* | PCR primers | 5'-GTTGATCTGTTCAGTGGTCTACA-3'; |
| Sequence-based reagent | cd59-F (for mutant genotyping) | This paper | PCR primers | 5'-TGGTAAACTGAAGGCATTATGAAA-3'; more information found in 'Generation of mutant lines' |
| Sequence-based reagent | cd59-R (for mutant genotyping) | This paper | PCR primers | 5'-GCAGGCATCATCATAGTAGCAG-3'; more information found in 'Generation of mutant lines' |
| Sequence-based reagent | cd59-F (for RT-PCR analysis) | This paper | PCR primers | 5'-ATGAAAGCTTCTGTCGGAGTGT-3'; more information found in 'RT-PCR analysis' |
| Sequence-based reagent | cd59-R (for RT-PCR analysis) | This paper | PCR primers | 5'-TTAGAAAACACCCCACCAGAAG-3'; more information found in 'RT-PCR analysis' |
| Sequence-based reagent | cd59 sgRNA-F (for sgRNA synthesis) | This paper | PCR primers | 5'-TAATACGACTCACTATAGGGCTGGCTCTGCTGGGGCTGTTTAGAGCTAGAAATAGCAAG-3'; more information found in 'Generation of mutant lines' |
| Sequence-based reagent | Constant oligonucleotide-R (for sgRNA synthesis) | *Gagnon et al., 2014* | PCR primers | 5'-AAAAGCACCGACTCGGTGCCACTTTTTCAAGTTGATAACGGACTAGCCTTATTTTAACTTGCTATTTCTAGCTCTAAAAC-3' |
| Sequence-based reagent | cd59-F (for CISH RNA probe) | This paper | PCR primers | 5'-GCCTGCTTGTCTGTCTACGA-3'; more information found in 'In situ hybridization' |
| Sequence-based reagent | cd59-R+T7 (for CISH RNA probe) | This paper | PCR primers | 5'-TAATACGACTCACTATAGAGGTGACGAGATTAGCTGCG-3'; more information found in 'In situ hybridization' |
| Software, algorithm | ImageJ/Fiji | | RRID:SCR_003070 | |

*Continued on next page*

*Continued*

| Reagent type (species) or resource | Designation | Source or reference | Identifiers | Additional information |
|---|---|---|---|---|
| Software, algorithm | Prism 9.2 | GraphPad Software | RRID:SCR_002798 | |
| Software, algorithm | Metamorph | Molecular Devices | RRID:SCR_002368 | |
| Software, algorithm | Andor iQ 3.6.3 | Oxford Instruments | RRID:SCR_014461 | |
| Software, algorithm | Imaris 9.8.0 | Oxford Instruments | RRID:SCR_007370 | |
| Software, algorithm | RStudio | RStudio | RRID:SCR_000432 | |
| Software, algorithm | Illustrator | Adobe | RRID:SCR_010279 | |
| Software, algorithm | CHOPCHOP | *Labun et al., 2019; Labun et al., 2016; Gagnon et al., 2014* | https://chopchop.cbu.uib.no/ | |
| Software, algorithm | CRISPRscan | *Moreno-Mateos et al., 2015* | https://www.crisprscan.org/ | |

## Zebrafish husbandry

All animal studies were approved by the University of Virginia Institutional Animal Care and Use Committee. Adult zebrafish were housed in tanks of 8–10 fish/L in 28.5°C water. Pairwise mating of adult zebrafish generated zebrafish embryos for all experiments. The embryos were raised in egg water (0.3 g instant ocean sea salt per L reverse osmosis water) contained in 10 cm Petri dishes and incubated at 28.5°C. Embryos used for experiments were staged by hpf or dpf (*Kimmel et al., 1995*). To minimize visual obstruction by pigmentation, egg water was exchanged for 0.004% 1-phenyl-2-thiourea (PTU; Sigma) in egg water at 24 hpf. Tricaine-S (MS-222; The Pond Outlet) was utilized as an anesthetic for embryos and larvae used in live imaging and euthanasia. Embryo and larvae sex were undetermined for all experiments because sex cannot be ascertained until ~25 dpf in zebrafish (*Takahashi, 1977*). To maintain genetic diversity, transgenic lines were renewed through outcrossing.

## Zebrafish transgenic lines

All transgene descriptions and abbreviations are included in the Key resources table. Transgenic lines produced during this study include *Tg(cd59(5.0):tagrfp)[uva52]* and *Tg(mbpa(6.6):tagrfp-caax)[uva53]*. The methods used to generate these lines are described in 'Generation of transgenic lines.' Previously published strains used in this study include AB*, *Tg(sox10(7.2):megfp)[sl3]* (*Kirby et al., 2006*), *Tg(sox10(4.9):tagrfp)[uva5]* (*Zhu et al., 2019*), *Gt(erbb3b:gal4);Tg(uas:egfp)* (*Brown et al., 2022*), *Tg(olig2:dsred2)[vu19]* (*Shin et al., 2003*), *Tg(sox10(4.9):nls-eos)[w18]* (*McGraw et al., 2012*), and *Tg(mbp(2.0):egfp-caax)[ue2]* (*Almeida et al., 2011*). All transgenic lines described are stable and incorporated into the germline. In addition to the stable lines, *cd59:tagrfp*, *mbpa:tagrfp-caax* and *mbp:egfp-caax* were also injected into embryos at the one-cell stage to create mosaic labeling (*Kawakami, 2004*).

## Zebrafish mutant lines

Mutant lines generated in the course of this study are as follows: *cd59[uva47]* (BX957297.10:g.45_50delGCTGGG) and *cd59[uva48]* (BX957297.10:g.44_58delTGCTGGGGCTTGGTA). The methods used to generate these lines are described in 'Generation of mutant lines.' Mutant and wildtype fish were distinguished with PCR amplification and gel electrophoresis of the mutant allele (see 'Generation of mutant lines'). Descriptions of all mutant lines and their abbreviations can be found Key resources table.

## Generation of transgenic lines

The Tol2kit Gateway-based cloning system was used to produce transgenic constructs (*Kwan et al., 2007*). The following entry and destination vectors were used in the creation of these constructs:

*p5E-cd59* (this article), *p5E-mbpa* (this article), *pME-tagrfp-caax* (*Auer et al., 2015*), *pME-tagrfp* (*Don et al., 2017*), *p3E-polya* (*Kwan et al., 2007*), *pDestTol2pA2* (*Kwan et al., 2007*), and *pDestTol-2pA2cryegfp*. These constructs were then used to produce the *Tg(cd59:tagrfp)* and *Tg(mbpa:tagrfp-caax)* zebrafish lines.

The *Tg(cd59(5.0):tagrfp)* line was generated as follows: PCR amplification of wildtype genomic DNA with forward primer, 5′-TCAGATCACATCACACCTGA-3′, and reverse primer, 5′-AATGCCTTCAGT TTACCAGTCT-3′, was used to clone 5 kb of the sequence upstream to the *cd59* gene (BX957297.10). The PCR product was purified through with the QIAquick gel extraction kit after gel electrophoresis (QIAGEN). The PCR product was subcloned into a pENTR 5′-TOPO vector (Invitrogen) to create the *p5E-cd59(–5.0)* vector. The resulting p5E-*cd59(5.0)* vector was transformed into chemically competent *Escherichia coli* for amplification, isolated with the QIAprep spin miniprep kit (QIAGEN), and Sanger sequenced to verify accurate assembly. All Sanger sequencing described in the article were conducted through GENEWIZ (Azenta Life Sciences; https://www.genewiz.com/en). LR reaction (*Ashton et al., 2012*) was used to ligate *p5E-cd59(5.0)* (this article), *pME-tagrfp* (*Don et al., 2017*), *p3E-polya* (*Kwan et al., 2007*), and *pDestTol2pA2* (*Kwan et al., 2007*). The resulting *cd59(5.0):tagrfp* expression vector was transformed into chemically competent *E. coli* for amplification, isolated with the QIAprep spin miniprep kit (QIAGEN), and Sanger sequenced to verify accurate assembly. The stable *Tg(cd59(5.0):-tagrfp)* line was established through co-microinjections of *cd59(5.0):tagrfp* expression vector (50 ng/µL) and *Tol2* transposase mRNA (20 ng/µL; mRNA synthesis described in 'Generation of synthetic mRNA') in one-cell stage embryos (*Kawakami, 2004*). Founders were screened for germline incorporation of the transgene.

Similarly, the *Tg(mbpa(6.6):tagrfp-caax)* line was as described for the *Tg(cd59(5.0):tagrfp)* line except: PCR amplification of wildtype genomic DNA with forward primer, 5′-ATGTCGAGTAATATCG AGCAGC-3′ (this article), and reverse primer, 5′-GTTGATCTGTTCAGTGGTCTACA-3′ (*Almeida et al., 2011*), was used to clone 6.59 kb of the sequence upstream to the *mbpa* gene (CU856623.7). The PCR product was subcloned into a pENTR 5′-TOPO vector (Invitrogen) to create the *p5E-mbpa(–6.6)* vector. LR reaction (*Ashton et al., 2012*) was used to ligate *p5E-mbpa(–6.6)* (this article), *pME-tagrfp-caax* (*Auer et al., 2015*), *p3E-polya* (*Kwan et al., 2007*), and *pDestTol2pA2cryegfp* (*Kwan et al., 2007*). The final *mbpa(6.6):tagrfp-caax* construct and stable zebrafish lines were generated as described for *cd59(5.0):tagrfp*.

## Generation of mutant lines

*cd59* mutant lines generated during the course of this study were produced using CRISPR/Cas9 genome editing according to the methods described in *Gagnon et al., 2014* except: the sgRNA targeting 5′-GTGCTGGCTCTGCTGGGGCTTGG-3′ in the second exon of *cd59* was identified with *CRISPRscan* (*Moreno-Mateos et al., 2015*) and *Chop Chop* (*Labun et al., 2016*; *Labun et al., 2019*; *Gagnon et al., 2014*) and synthesized through PCR amplification with the following primers: forward primer, 5′-taatacgactcactataGGGCTGGCTCTGCTGGGGCTgttttagagctagaaATAGCAAG-3′, and constant oligonucleotide reverse primer, 5′-AAAAGCACCGACTCGGTGCCACTTTTTCAAGTTGAT AACGGACTAGCCTTATTTTAACTTGCTATTTCTAGCTCTAAAAC-3′ (*Gagnon et al., 2014*). All sgRNA synthesis and cleanup were performed as described in *Gagnon et al., 2014*.

To generate a mutation in *cd59*, the sgRNA (200 ng/µL) was co-microinjected with Cas9 protein (600 ng/µL; PNA Bio) into embryos at the one-cell stage. Successful insertion and deletion mutation (INDEL) generation was verified according to the *Gagnon et al., 2014* protocol using the following primers: forward primer, 5′-TGGTAAACTGAAGGCATTATGAAA-3′, and reverse primer, 5′-GCAGGCAT CATCATAGTAGCAG-3′.

To establish stable mutant lines, injected embryos were raised to adulthood and outcrossed with wildtype zebrafish. Founder offspring were screened for germline mutations through PCR amplification of *cd59* with the same genotyping primers listed above. To identify mutant alleles, the resulting PCR product was cloned into pCR4-TOPO TA vector (Invitrogen). Vectors containing *cd59* mutant alleles were amplified in chemically competent *E. coli*, isolated through miniprep (QIAGEN), and sequences were evaluated for INDELs. Founder offspring containing 6 bp (*cd59*[uva47]; BX957297.10:g.45_50del-GCTGGG) and 15 bp deletions (*cd59*[uva48]; BX957297.10:g.44_58delTGCTGGGGCTTGGTA) in the *cd59* gene were selected for further experimentation and raised to establish stable mutant lines. All subsequent generations were genotyped as described in *Fontenas and Kucenas, 2021* with the same

genotyping primers listed above, and the resulting PCR products were screen with gel electrophoresis on a 2.5% agarose gel.

## Generation of synthetic mRNA

To aid in transgenic fish creation, *Tol2* transposase mRNA was transcribed from linearized pCS2FA-transposase (*Kwan et al., 2007*) with the mMESSAGE mMACHINE sp6 transcription kit (Invitrogen). The resulting mRNA was co-injected (20 ng/μL) with the transgenic constructs described in 'Generation of transgenic lines'.

## RT-PCR analysis

RNA was extracted with the RNeasy mini kit (QIAGEN) from 72 hpf *cd59*$^{uva48}$, cd59$^{uva47}$, and AB* embryos. cDNA libraries were generated from the RNA using the high-capacity cDNA reverse transcription kit (Thermo Fisher). Using the cDNA as a template, PCR amplification was performed with the following primers: forward primer, 5′-ATGAAAGCTTCTGTCGGAGTGT-3′, reverse primer, 5′-ATGA AAGCTTCTGTCGGAGTGT-3′. The resulting PCR products were visualized with gel electrophoresis on a 2.5% agarose gel.

To sequence the multiple mutant RNA transcripts found in the *cd59*$^{uva48}$ mutant, the PCR product was cloned into a pCR4-TOPO TA cloning vector (Invitrogen) as described in 'Generation of mutant lines.' The isolated vectors were sequenced and analyzed for INDELs and PTCs with the Expasy protein translation tool (https://web.expasy.org/translate/; *Duvaud et al., 2021*). *Cd59*$^{uva48}$ protein sequences were compared to sequences from *cd59*$^{uva47}$ and AB* larvae.

## Confocal imaging

All embryos were treated with egg water containing PTU (0.004%; Sigma) at 24 hpf to minimize pigmentation obstruction. For in vivo imaging, embryos and larvae (1–7 dpf) were dechorionated manually, if necessary, and anesthetized with 0.01% tricaine-S (MS-222) (The Pond Outlet). Low gelling temperature agarose (0.8%; Sigma) was used to immobilize the anesthetized fish in a 35 mm glass-bottom dish (Greiner). Egg water containing PTU (0.004%; Sigma) and tricaine-S (MS-222) (0.01%; The Pond Outlet) was added to the dish prior to imaging to maintain anesthesia and suppress pigment production. For whole-mount imaging of fixed fish (see 'Immunofluorescence'), fixed embryos and larvae were immobilized with agarose in a glass-bottom dish prior to imaging. For imaging tissue sections, sections were adhered to microscope slides prior to staining and imaging (see 'Cryosectioning' and 'Immunofluorescence').

All fluorescent images were acquired with a ×40 water immersion objective (NA = 1.1) mounted on a motorized Zeiss AxioObserver Z1 microscope equipped with a Quorum WaveFX-XI (Quorum Technologies) or Andor CSU-W (Andor Oxford Instruments plc.) spinning disc confocal system. Time-lapse experiments were imaged every 10 min for 7–24 hr, depending on the experiment. Z stacks were acquired at each time point for time-lapse imaging as well as single-time point imaging for fixed whole-mount and sectioned tissue (see 'Immunofluorescence,' 'In situ hybridization,' and 'Cryosectioning').

All experiments involving whole embryos and larvae were imaged with the 12th somite at the center of the acquisition window to control for stage of anterior–posterior development. Exceptions include (1) images of mosaic labeling with transgenic constructs, which were acquired regardless of anterior–posterior position, (2) images of Na$_V$ channels and NF183, which were acquired from the 3rd to the 13th somite to ensure accurate assessment of nodes of Ranvier that are not evenly distributed along nerves, (3) images of the pLLG, which is anterior to the pLLN. All imaging of mosaic labeling as well as myelin volume quantification were obtained in live fish.

Images and videos were processed with either Metamorph (Molecular Devices) or IQ3 (Oxford Instruments). Fiji (ImageJ; imageJ.nih.gov), Imaris 9.8 (Oxford Instruments), and Illustrator (Adobe) were used for annotating videos and images, adjusting contrast and brightness, and data analysis.

## Cryosectioning

Fixed larvae were mounted in sectioning agar (1.5% agar; Fisher Scientific; 5% sucrose; Sigma; 100 mL ultrapure water) and cryopreserved in 30% sucrose in ultrapure water (Sigma) overnight at 4°C. The agar blocks were frozen by placing them on a small raft floating on 2-methylbutane (Fisher Scientific), the container of which was submerged in a bath of liquid nitrogen. The blocks were sectioned to

20 µm with a cryostat microtome and mounted on microscope slides (VWR). The sections were stored at –20°C until needed for immunofluorescence, ISH, or confocal imaging.

Prior to sectioning the offspring of heterozygous *cd59* mutant parents, the larvae were anesthetized in egg water with 0.01% tricaine-s (MS-222; The Pond Outlet). The heads were removed with a razor blade. The heads were kept for genotyping, and the trunks were fixed for sectioning.

## Immunofluorescence

Larvae (24 hpf to 7 dpf) were fixed in 4% paraformaldehyde (PFA; Sigma) for 1 hr shaking at room temperature (RT) for all experiments except $Na_V$ channel and NF186 staining, in which samples were fixed for 30 min.

For whole-mount imaging of embryo and larvae, the samples were prepared as described in *Fontenas and Kucenas, 2021*. The following antibodies were used for whole-mount immunofluorescence staining: mouse anti-acetylated tubulin (1:10,000; Sigma), rabbit anti-sox10 (1:5000) (*Binari et al., 2013*), chicken anti-GFP (1:500; Abcam), mouse anti-sodium channel (1:500; Sigma), rabbit anti-TagRFP (1:500; Invitrogen), rabbit anti-neurofascin 186 (1:200; gift from Dr. Matthew Rasband), mouse anti-C5b-8+C5b-9 (1:500; Abcam), Alexa Fluor 488 goat anti-chicken IgG(H+L) (1:1000; Thermo Fisher), Alexa Fluor 647 goat anti-rabbit IgG(H+L) (1:1000; Thermo Fisher), Alexa Fluor 647 goat anti-mouse IgG(H+L) (1:1000; Thermo Fisher), Alexa Fluor 488 goat anti-mouse IgG(H+L) (1:1000; Thermo Fisher), and Alexa Fluor 568 goat anti-rabbit IgG(H+L) (1:1000; Thermo Fisher). Fish were immobilized and imaged in glass-bottom dishes as described in 'Confocal imaging'.

For imaging of tissue sections, the samples were prepared as described in *Fontenas and Kucenas, 2021*. The following antibodies were used for staining tissue sections: mouse anti-HuC/HuD (1:500; Invitrogen), rabbit anti-sox10 (1:5000) (*Binari et al., 2013*), Alexa Fluor 488 goat anti-mouse IgG(H+L) (1:1000; Thermo Fisher), and Alexa Fluor 647 goat anti-rabbit IgG(H+L) (1:1000; Thermo Fisher). After staining, all slides were mounted in DAPI fluoromount-G (Southern Biotech) and were coverslipped (VWR). The slides were stored in the dark until they were imaged as described in 'Confocal imaging'.

## In situ hybridization

### Probe synthesis

The *cd59* RNA probe for CISH was designed and synthesized in our lab. RNA was isolated from 3 dpf larvae with the RNeasy mini kit (QIAGEN). A cDNA library was generated from the 3 dpf RNA using the high-capacity cDNA reverse transcription kit (Thermo Fisher). This cDNA was used as a template for the *cd59* RNA probe. The primers used to generate the *cd59* RNA probe were as follows: forward primer, 5'-GCCTGCTTGTCTGTCTACGA-3', and reverse primer plus T7 sequence, 5'-TAAT ACGACTCACTATAGAGGTGACGAGATTAGCTGCG-3'. In addition to the *cd59* RNA probe, we also used previously published probes targeting *sox10* (*Park and Appel, 2003*) and *mbpa* (*Brösamle and Halpern, 2002*). CISH probe synthesis was performed as described in *Fontenas and Kucenas, 2021*. For fluorescent in situ hybridization experiments (FISH), the *cd59* RNA probe (RNAscope probe-Dr-cd59-C2) was purchased from Advanced Cell Diagnostics (ACD).

### Chromogenic in situ hybridization

Embryos and larvae (1–7 dpf) were dechorionated, if necessary, and fixed in 4% PFA for 1 hr shaking at RT and then transferred to 100% MeOH overnight at –20°C. CISH was performed as described in *Hauptmann and Gerster, 2000*. Images were obtained either using a Zeiss AxioObserver inverted microscope equipped with Zen, using a ×40 oil immersion objective, or a Zeiss AxioObserver Z1 microscope equipped with a Quorum WaveFX-XI (Quorum Technologies) or Andor CSU-W (Andor Oxford Instruments plc.) spinning disc confocal system. Fiji (ImageJ; imageJ.nih.gov) and Illustrator (Adobe) were used for annotating images, adjusting contrast and brightness, and data analysis.

### Fluorescent in situ hybridization

Larvae (1–7 dpf) were dechorionated, if necessary, and fixed with 4% PFA (Sigma) for 1 hr shaking at RT, dehydrated in 100% MeOH overnight at –20°C, and cryosectioned (see 'Cryosectioning'). The agar was gently removed by soaking the slides in 1× DEPC PBS (1:1000 diethyl pyrocarbonate [DEPC; Sigma] in 1× PBS; DEPC inactivated by autoclave).

To perform FISH, we used the RNAscope fluorescent multiplex reagent kit (ACD) with the following protocol modified from *Gross-Thebing et al., 2014*: for all of subsequent steps, the slides kept in the dark and were coverslipped (VWR) during incubations except for during washes. To permeabilize the tissue, the sections were treated with Protease III (two drops) and incubated at RT for 20 min. The tissue was gently rinsed three times with 1× DEPC PBS and washed with 1× DEPC PBS for 10 min at RT followed by an addition three rinses in 1× DEPC PBS. The sections were then hybridized with *cd59* probe (1:100 RNAscope probe-Dr-cd59-C2 in RNAscope probe diluent; ACD) in a 40°C water bath overnight. The tissue was rinsed three times with 1× DEPC PBS and washed in 0.2× SSCT (0.2× SSC [Quality Biological] and 0.1% Triton X-100 [Sigma] in DEPC water [1:1000 DEPC in ultrapure water; DEPC inactivated by autoclave]) for 10 min and then rinsed again three times with 1× DEPC PBS. This rinse/wash routine was repeated between the following fixation and amplification steps: 4% PFA (5 min at RT), Amp1 (two drops, 30 min at 40°C), Amp2 (two drops, 15 min at 40°C), Amp3 (two drops, 30 min at 40°C), Amp4C (two drops, 15 min at 40°C), and DAPI (two drops, 30 min at RT). After a final rinse/wash, the slides were coverslipped (VWR) with 0.2× SSCT and stored at 4°C until imaged. Sections were imaged as described in 'Confocal imaging'.

## Cell death assays

### AO incorporation assay
To label cell death, 48 hpf embryos were labeled with AO according to the protocol in *Lyons et al., 2005*. Fish were immobilized and imaged in glass-bottom dishes as described in 'Confocal imaging'.

### TUNEL assay
To label cell death, 48 hpf embryos were fixed in 4% PFA for 1 hr shaking at RT and then sectioned (see 'Cryosectioning'). TUNEL staining was performed with the ApopTag red in situ apoptosis detection kit (Sigma) and followed by immunofluorescence for Sox10 (see 'Immunofluorescence'). Sections were imaged as described in 'Confocal imaging'.

## EdU incorporation assay
To label mitotically active cells, embryos were incubated in EdU (0.4 mM EdU [Invitrogen], 4% DMSO [Sigma], and egg water containing 0.004% PTU [Sigma]) from 48 to 55 hpf. The embryos were then fixed in 4% PFA for 1 hr shaking at RT and permeabilized as described in 'Immunofluorescence.' We stained for EdU with the Click-it EdU cell proliferation kit for imaging (Alexa Fluor 647 dye; Invitrogen) as described in the kit protocol. The Click-it reaction was performed for 1 hr shaking at RT. The embryos were washed overnight shaking at 4°C in 1× PBSTx. Additional washes were performed as necessary until the yolk sacs were no longer blue. Fish were immobilized and imaged in glass-bottom dishes as described in 'Confocal imaging'.

## Dexamethasone treatment
To inhibit inflammation, embryos were incubated in 1% DMSO (Sigma) or 1% DMSO plus 100 µM Dex (Sigma) at 28.5°C. Dex concentration was chosen based on a previously published dose–response study in zebrafish larvae (*Wilson et al., 2016*; *Wilson et al., 2013*). For embryos fixed at 55 hpf, the embryos were treated with DMSO or Dex from 24 to 55 hpf. For larvae fixed at 7 dpf, the embryos were treated from 24 to 75 hpf and then transferred to egg water containing PTU (0.004%; Sigma).

## Transmission electron microscopy
Headless larvae (7 dpf) were fixed for 3 days at 4°C in EM fixation buffer (2% EM-grade glutaraldehyde [Sigma], 2% PFA [Sigma], 0.1 M sodium cacodylate trihydrate, pH 7.3 [SCT; Sigma] in ultrapure water) and washed three times for 10 min each in 0.1 M SCT. The samples were post-fixed in 1% osmium tetroxide (Sigma) in 0.1 M SCT for 1 hr at 4°C and washed three times for 5 min each in ultrapure water at RT. Contrast was initiated with 2% uranyl acetate for 1 hr at RT and washed four times for 5 min each in ultrapure water at RT. The samples were dehydrated as follows: 40% EtOH (2 × 10 min), 60% (2 × 10 min), 80% (2 × 10 min), 100% (2 × 10 min), acetone (2 min), and acetone/EPON (1:1, 15 min). The samples were then incubated in EPON at 4°C overnight with the lid open to allow for gas to escape. Samples were mounted in EPON and polymerized at 60°C for 48 hr. Tissue blocks were sectioned transversely, stained, and imaged according to the methods described in *Morris et al.,*

*2017*. Sections were collected from three parts of the larvae, separated by 100 µm (according to the myelin length data noted in *Figure 5—figure supplement 1D*, most myelin sheaths are less than 100 µm; separating each section by 100 µm would therefore enable quantification of three separate groups of SCs per pLLN) to enable quantification of three groups of SCs per pLLN.

## RNAseq analysis

Published bulk and scRNAseq datasets from zebrafish and rodents were evaluated for *cd59* expression. The chosen datasets included analysis of some or many stages of myelinating glial cell development. Visualization and quantification of *cd59* expression were obtained through the applications included with the publications when possible (publications cited in *Figure 1—figure supplement 1*) with the following exceptions: (1) TPM quantification of bulk RNAseq of zebrafish myelinating glial cells was acquired from data analyzed in *Piller et al., 2021*; *Zhu et al., 2019*. (2) analysis of NCC lineage scRNAseq in zebrafish from the dataset presented in *Saunders et al., 2019* was performed as follows: tissue collection, FACS, RT-PCR, single-cell collection, library construction, sequencing, read alignment to Ensembl GRCz11, and preliminary data processing were performed as described by *Saunders et al., 2019*, and the data were accessed through GEO via accession GSE131136. Dimensionality reduction and projection of cells in two dimensions were performed using uniform manifold approximation and projection (UMAP) (*McInnes et al., 2018*) and Louvain clustering (*Blondel et al., 2008*). Cell cluster visualization and analysis were performed using the FindMarkers, FeaturePlot, DimPlot, and VlnPlot functions of Seurat package 4.0.0 (*Hao et al., 2021*).

## Quantification and statistical analysis

### Nerve volume quantification

Myelinated nerve and axon volume were quantified by creating a surface rendering of the imaged nerve using Imaris 9.8 (Oxford Instruments). The total volume of each surface rendering was quantified with Imaris 9.8 (Oxford Instruments).

### Myelin length quantification

Myelin length was measured from images of mosaic *mbpa:tagrfp-caax* labeling using Fiji (ImageJ; imageJ.nih.gov). Specifically, the length of the myelin sheath was traced with the 'freehand line' tool. The length was then quantified with the 'measure' tool.

### Axon myelination quantification

The number of myelin wraps around an axon, as well as the number of myelinated axons per pLLN, was measured in electron micrographs of pLLNs using Fiji (ImageJ; imageJ.nih.gov). Counting was performed with the 'multi-point' and 'measure' tools.

### Na$_V$ channel, NF186, cd59 RNAscope, and MAC quantification

Puncta quantification (specifically for quantification of Na$_V$ channels, NF186, *cd59* RNAscope, and MAC quantification) was performed either in Fiji (ImageJ; imageJ.nih.gov) or Imaris (Oxford Instruments). To ensure only puncta within the pLLN were quantified, each z plane was quantified manually.

### Cell number quantification

SC, neuron, and NCC images on the pLLN and in the CNS were quantified in Fiji (ImageJ; imageJ.nih.gov) by creating z projections and counting the number of cells with the 'multi-point' and 'measure' tools. pLLG neurons and NCCs were counted manually in Imaris 9.8 (Oxford Instruments).

### Mitotic event quantification

Time-lapse images of developing SCs were evaluated for mitotic events (cell divisions) of *sox10:tagrfp*-positive SCs along the pLLN using Fiji (ImageJ; imageJ.nih.gov). Mitotic events were defined as follows: SC rounds up, chromatic condenses (darkening of the SC), and division of the daughter cells. The number of mitotic events was quantified with the 'multi-point' and 'measure' tools. Mitotic events in the representative videos were annotated arrows using the 'draw arrow in movies' plugin (*Daetwyler et al., 2020*).

## Statistical analysis

Student's *t*-tests, as well as one-way and two-way ANOVAs followed by Tukey's multiple-comparison tests, were performed using Prism 9.2 (GraphPad Software). The data in plots and the text are presented as means ± SEM.

## Data blinding

For offspring of heterozygous parents, embryos chosen for experimentation were blindly selected (wildtype and mutant embryos are indistinguishable by eye). Embryo genotype was revealed after imaging and before quantification. For offspring of homozygous parents, embryo genotype was known throughout the entire experiment.

## Artwork

All artworks were created in Illustrator (Adobe) by Ashtyn T. Wiltbank. SC and OL development figures (*Figure 1A and B*) were based on previous schematics and electron micrographs shown in *Ackerman and Monk, 2016*; *Cunningham and Monk, 2018*; *Jessen and Mirsky, 2005*.

## Acknowledgements

We thank Lori Tocke for zebrafish care and members of the Kucenas Lab for valuable discussions. We thank Drs. Laura Fontenas, Xiaowei Lu, Sarah E Siegrist, John R Lukens, and Scott Zeitlin for helpful suggestions and training. We thank Drs. Laura Fontenas, Alev Erisir, Stacey J Criswell, Natalia Dworak, and the Advanced Microscopy Facility for assistance with EM sample preparation and microscope training. We thank Dr. Matthew N Rasband for the NF186 antibody. We thank Dr. David M Parichy for his advice and for sharing scRNAseq data from (*Saunders et al., 2019*). This work was funded by the NIH/National Institutes of Neurological Disorders and Stroke (NINDS): R01NS107525 (SK) and The Owens Family Foundation (SK).

# Additional information

## Funding

| Funder | Grant reference number | Author |
| --- | --- | --- |
| National Institute of Neurological Disorders and Stroke | R01NS107525 | Sarah Kucenas |
| Owens Family Foundation | | Sarah Kucenas |

The funders had no role in study design, data collection and interpretation, or the decision to submit the work for publication.

## Author contributions

Ashtyn T Wiltbank, Conceptualization, Data curation, Formal analysis, Investigation, Methodology, Project administration, Resources, Validation, Visualization, Writing – original draft, Writing – review and editing; Emma R Steinson, Formal analysis, Investigation, Validation; Stacey J Criswell, Methodology; Melanie Piller, Data curation, Formal analysis, Visualization, Writing – original draft; Sarah Kucenas, Conceptualization, Funding acquisition, Project administration, Supervision, Validation, Writing – original draft, Writing – review and editing

## Author ORCIDs

Sarah Kucenas (ID) http://orcid.org/0000-0002-1950-751X

## Ethics

All animal studies were approved by the University of Virginia Institutional Animal Care and Use Committee, protocol #3782.

Decision letter and Author response
Decision letter https://doi.org/10.7554/eLife.76640.sa1
Author response https://doi.org/10.7554/eLife.76640.sa2

# Additional files

## Supplementary files
• Transparent reporting form

## Data availability
All data generated and analyzed during this study are included in the manuscript and supporting source files for Figures 1 to 8. All previously published datasets analyzed, including RNAseq and proteomics, are cited in the manuscript.

The following previously published datasets were used:

| Author(s) | Year | Dataset title | Dataset URL | Database and Identifier |
|---|---|---|---|---|
| Howard AGA, Baker PA, Ibarra-García-Padilla R, Moore JA, Rivas LJ, Tallman JJ, Singleton EW, Westheimer JL, Corteguera JA, Uribe RA | 2021 | An atlas of neural crest lineages along the posterior developing zebrafish at single-cell resolution | https://www.ncbi.nlm.nih.gov/geo/query/acc.cgi?acc=GSE152906 | NCBI Gene Expression Omnibus, GSE152906 |
| Zhu Y, Crowley SC, Latimer AJ, Lewis GM, Nash R, Kucenas S | 2019 | Migratory Neural Crest Cells Phagocytose Dead Cells in the Developing Nervous System | https://www.ncbi.nlm.nih.gov/geo/query/acc.cgi?acc=GSE135237 | NCBI Gene Expression Omnibus, GSE135237 |
| Gerber D, Pereira JA, Gerber J, Tan G, Dimitrieva S, Yánguez E, Suter U | 2021 | Transcriptional profiling of mouse peripheral nerves to the single-cell level to build a sciatic nerve ATlas (SNAT) | https://www.ncbi.nlm.nih.gov/geo/query/acc.cgi?acc=GSE137870 | NCBI Gene Expression Omnibus, GSE137870 |
| Marisca R, Hoche T, Agirre E, Hoodless LJ, Barkey W, Auer F, Castelo-Branco G, Czopka T | 2020 | Functionally distinct subgroups of oligodendrocyte precursor cells integrate neural activity and execute myelin formation | https://www.ncbi.nlm.nih.gov/geo/query/acc.cgi?acc=GSE132166 | NCBI Gene Expression Omnibus, GSE132166 |
| Marques S, Zeisel A, Codeluppi S, Bruggen D, Falcão AM, Xiao L, Li H, Häring M, Hochgerner H, Romanov RA, Gyllborg D, Muñoz-Manchado AB, Manno GL, Lönnerberg P, Floriddia EM, Rezayee F, Ernfors P, Arenas E, Hjerling-Leffler J, Harkany T, Richardson WD, Linnarsson S, Castelo-Branco G | 2016 | Oligodendrocyte heterogeneity in the mouse juvenile and adult central nervous system | https://www.ncbi.nlm.nih.gov/geo/query/acc.cgi?acc=GSE75330 | NCBI Gene Expression Omnibus, GSE75330 |

*Continued on next page*

*Continued*

| Author(s) | Year | Dataset title | Dataset URL | Database and Identifier |
|---|---|---|---|---|
| Marques S, Bruggen D, Vanichkina DP, Floriddia EM, Munguba H, Väremo L, Giacomello S, Falcão AM, Meijer M, Björklund ÅK, Hjerling-Leffler J, Taft RJ, Castelo-Branco G | 2018 | Transcriptional Convergence of Oligodendrocyte Lineage Progenitors during Development | https://www.ncbi.nlm.nih.gov/geo/query/acc.cgi?acc=GSE95194 | NCBI Gene Expression Omnibus, GSE95194 |
| Piller M, Werkman IL, Brown EA, Latimer AJ, Kucenas S | 2021 | Glutamate Signaling via the AMPAR Subunit GluR4 Regulates Oligodendrocyte Progenitor Cell Migration in the Developing Spinal Cord | https://www.ncbi.nlm.nih.gov/geo/query/acc.cgi?acc=GSE174486 | NCBI Gene Expression Omnibus, GSE174486 |
| Saunders LM, Mishra AK, Aman AJ, Lewis VM, Toomey MB, Packer JS, Qiu X, McFaline-Figueroa JL, Corbo JC, Trapnell C, Parichy DM | 2019 | Thyroid hormone regulates distinct paths to maturation in pigment cell lineages | https://www.ncbi.nlm.nih.gov/geo/query/acc.cgi?acc=GSE131136 | NCBI Gene Expression Omnibus, GSE131136 |
| Wolbert J, Li X, Heming M, Mausberg AK, Akkermann D, Frydrychowicz C, Fledrich R, Groeneweg L, Schulz C, Stettner M, Gonzalez NA, Wiendl H | 2020 | Redefining the heterogeneity of peripheral nerve cells in health and autoimmunity | https://www.ncbi.nlm.nih.gov/geo/query/acc.cgi?acc=GSE142541 | NCBI Gene Expression Omnibus, GSE142541 |
| Siems SB, Jahn O, Hoodless LJ, Jung RB, Hesse D, Möbius W, Czopka T, Werner HB | 2021 | Proteome Profile of Myelin in the Zebrafish Brain | http://proteomecentral.proteomexchange.org/cgi/GetDataset?ID=PXD023037 | ProteomeXchange, PXD023037 |

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
