## [Editor Report]

This work investigates the function of the small GPI-anchored protein Cd59, a protein known to suppress complement-mediated inflammation, in Schwann cell development and myelination, using zebrafish as a model system. This article will be of interest to developmental biologists, glial biologists, and immunologists as it suggests the interesting and novel findings that Cd59 regulates Schwann cell development, mainly by modulating Schwann cell proliferation.

---

## [Decision Letter]

**Decision letter after peer review:**

Thank you for submitting your article "Cd59 and inflammation orchestrate Schwann cell development" for consideration by *eLife*. Your article has been reviewed by 3 peer reviewers, and the evaluation has been overseen by a Reviewing Editor and Didier Stainier as the Senior Editor. The reviewers have opted to remain anonymous.

Essential revisions:

There is strong consensus by all three reviewers on the potential interest and novelty of the paper. However, reviewers 1 and 3 felt that more experiments are needed to support the major conclusions of the paper. I do agree with these reviewers that the data in figure 2, 3 and 5 should be significantly improved to be more convincing. I also agree that, while the rescue is impressive, desametazone has very broad action and more data are needed to corroborate that excessive proliferation and myelination were actually rescued after desametazone treatment. Although the suggestion of reviewer 1 to confirm the rescue using available drugs with a more restricted complement effect is valuable, I think that it would be sufficient to discuss the shortcoming of desametazone and tune down the claims about the effect of inflammation in the text and title. Reviewers 1 and 3 also had questions about the mechanisms, that relate to explanation regarding the large effect of Cd59 deletion, given the small percentage of cells expressing CD59, and also on how to explain the relationship between Schwann cell proliferation and impaired myelination or formation of nodes of Ranvier. I would add to this that it is very surprising that increased proliferation in Schwann cell precursors does not lead to increased apoptosis, as it is well known that Schwann cell precursors depend on axonal contact for survival.

*Reviewer #1 (Recommendations for the authors):*

1. What could impair myelination if Schwann cells are not overproliferating anymore at the time of myelination? Analysis of the Schwann cell cycle at the time of myelination could provide some explanation. We need at least expression data of the inducers and inhibitors of myelination. There is also no functional analyses. The impaired myelination and node of Ranvier formation must lead to functional impairment.

2. Figure 2—figure supplement 1, panel C: even with a lot of effort, I cannot see co-localization of cd59 (cyan) with Sox10:megfp (orange) at any time-point. It could be that some of the Sox10:megfp-labeled cells are cd59-positive, but arrows are missing on the orange channel and it is not clear whether those are confocal images or not. Whether cd59 is expressed in Sox10-expressing cells is an essential point for the study, so it is important to show convincing images. Co-localization is also not convincing in Figure 2B. In addition, the arrows seem to indicate cd59-positive cells and not Sox10:megfp SCs or OLs, as stated in the figure legend. Co-IF showing cd59:tagrfp expression in Sox10-positive SCs is more convincing. Please, add co-IF images at 3 or 4 dpf to illustrate quantification shown in Figure 2E. According to the Material and methods section, the IF images seem to have been taken by confocal microscopy. Please, add this information in the figure legend.

3. Figure 2¬—figure supplement 2: in panel A, co-localization is again not obvious at all. In panels B-D, what are we looking at? IF or in vivo labeling? Could the authors quickly clarify the expected subcellular localization of the different egfp reporters used here to avoid the need to look for this information in the referenced articles?

4. Figure 3G: same comment as above, not convincing. I really do not understand what we are looking at. In Figure 3H, the authors show a graph representing the number of puncta per cell. Does it mean that what is shown in Figure 3G is a single cell? To me, this does not look like a single cell. How do the authors define the limits of single cells? They need to show images with several cells so that we have an idea of how straightforward it is to detect the limits of each cell. In the figure legend of Figure 3G, it is not written that the images presented show single cells.

5. Figure 5A: THE EM images of the mutant and the control look very different in quality. Could the authors show images of comparable quality/resolution? Looking at these images, I wonder how easy it is to count individual wraps in a reliable manner. In Figure 5D, it seems that the total number of axons has been used as n for statistics and not the average of all axons per fish with a n of 3, as stated in the figure legend. For statistical analyses, the n should be the number of fish, n=3. If the n is too high, there is a risk of finding a significant difference where there is no real difference. Please, re-calculate in case the n used was indeed the number of axons and not the number of fish.

6. Figure 5C: The NF186 staining is not convincing at all. This needs to be improved. At the moment, it appears very unspecific. Co-localization with another nodal component or another antibody with a less non-specific signal could help. In addition, the resolution of the images shown in Figure 5B is not good enough and should also be improved. The mbpa:tagrfp-caax does not help much either. It would be better to use an antibody to detect MBP or Neurofilament, for example.

7. Figure 6A: Again, the staining here for C5b-9/C5b-8 is not convincing. First, the signal is very low and we don't see much. Second, it is not clear whether the authors permeabilized the tissues or not (this information does not seem to be given in the Material and methods section). C5b-9/C5b-8 should be localized at the cell surface, so a cross-view should be added to the longitudinal view to demonstrate the membrane localization.

8. Discussion: This part does not really discuss in depth the main claimed findings of the study and does not discuss the potential implication of the new findings in relation to the CD59 human deficiency and neurological dysfunctions resistant to treatment by complement inhibitors. This part thus appears incomplete.

*Reviewer #2 (Recommendations for the authors):*

This is an interesting and relevant manuscript and nice work, and everyone in the field should benefit from reading it. My congratulations! I have just a few observations that the authors may want to consider for their revision

-As CD59 is only one out of several factors affecting Schwann cell development (GPCRs, ECM, transcription factors,.…), I feel phrasing that CD59 'orchestrates' SC development (in the title) may be a little too strong. 'Regulates', maybe?

-Some letters and numbers are too small to be deciphered in Figure 1 Suppl 1 A, C1, C2 and D

-Image quality is comparatively poor in Figure 1 Sup1 A, B1, B2, B3. This may be just in my PDF copy, but if not the authors may want to search for ways to improve them.

-In Figure 1 Sup1 D legend, I suppose the authors mean 'myelinating OLCs', not 'mouse OLCs'

-Page 26, Line 226-227, I suppose the authors mean 117 AA, not 177 AA

- In case suitable samples and antibodies are available, Figure 3 would benefit from Western blot validation of the lack of CD59 in the zebrafish line.

-Figure 4 F, the authors may want to consider if these are truly representative images. The difference is not easily spotted.

-Why not switch Figure 4 S2 and Figure 4 S1? It may possibly fit better with the line of arguments. Not essential, just a suggestion.

-Figure 5 A, can the authors please double-check if these are truly representative images? I am asking because the axon diameters seem to differ in the mutants as judged by the example images. Are there enough suitable samples to quantify axonal diameters? Is it possible to say if axons are normally sorted out of bundles, i.e. to quantify the numbers and diameters of axons in bundles? Sorting defects are common pathology in the PNS, at least in mammals.

- In case suitable samples and antibodies are available, Fig6 would benefit from a Western blot for the complement. Not an essential revision

-Related to Fig7, in case suitable samples are available it would be interesting to see TEM of myelinated axons and nodes after Dexamethasone treatment. Not an essential revision, just interesting

- In case the required samples are available, it would be interesting to see some examination of adult WT vs mutant zebrafish. Not an essential revision, again.

---

## [Author Response]

Essential revisions:There is strong consensus by all three reviewers on the potential interest and novelty of the paper. However, reviewers 1 and 3 felt that more experiments are needed to support the major conclusions of the paper. I do agree with these reviewers that the data in figure 2, 3 and 5 should be significantly improved to be more convincing. I also agree that, while the rescue is impressive, desametazone has very broad action and more data are needed to corroborate that excessive proliferation and myelination were actually rescued after desametazone treatment. Although the suggestion of reviewer 1 to confirm the rescue using available drugs with a more restricted complement effect is valuable, I think that it would be sufficient to discuss the shortcoming of desametazone and tune down the claims about the effect of inflammation in the text and title. Reviewers 1 and 3 also had questions about the mechanisms, that relate to explanation regarding the large effect of Cd59 deletion, given the small percentage of cells expressing CD59, and also on how to explain the relationship between Schwann cell proliferation and impaired myelination or formation of nodes of Ranvier. I would add to this that it is very surprising that increased proliferation in Schwann cell precursors does not lead to increased apoptosis, as it is well known that Schwann cell precursors depend on axonal contact for survival.

Thank you for your comments. We have added data to improve figures 2, 3, and 5 as well as expanded our discussion to speculate on some of the potential mechanisms at play and other interesting questions elicited by the results of this study. We have also adjusted our claims to reflect the shortcomings of dexamethasone. We have addressed the rest of the reviewer comments below.

With regards to the SCPs, it appears that there is enough room on the nerve to accommodate the extra cells. From our time-lapse imaging, we observe that all SCPs have contact with the nerve, which we speculate must be sufficient to maintain their survival.

Reviewer #1 (Recommendations for the authors):1. What could impair myelination if Schwann cells are not overproliferating anymore at the time of myelination? Analysis of the Schwann cell cycle at the time of myelination could provide some explanation. We need at least expression data of the inducers and inhibitors of myelination. There is also no functional analyses. The impaired myelination and node of Ranvier formation must lead to functional impairment.

Thank you for your comment. We added discussion around potential mechanisms for impaired myelination in our text.

In addition to the *sox10* and *mbp* expression we show in Figure 3E and F, we have added analysis of *krox20* expression. These data indicate that the *cd59* mutant SCs are expressing genes that are necessary for SC development and myelination. In addition to these data, the *cd59* mutant SCs also produce myelin at the correct developmental stage, indicating that these cells are capable of myelination.

Given these data, we propose two possible hypotheses in our discussion as to why myelin formation is reduced in the cd59 mutants.

With regards to functional impairment, we agree that this will be interesting to look at in future studies. At the developmental stages we are looking at, it is not typical to see functional issues and therefore, we are not surprised that there is no obvious phenotype. However, in a future study, we would love to look closer at the consequences of these myelin and node of Ranvier defects in our adult mutants, which do appear to have neurological dysfunction.

2. Figure 2—figure supplement 1, panel C: even with a lot of effort, I cannot see co-localization of cd59 (cyan) with Sox10:megfp (orange) at any time-point. It could be that some of the Sox10:megfp-labeled cells are cd59-positive, but arrows are missing on the orange channel and it is not clear whether those are confocal images or not. Whether cd59 is expressed in Sox10-expressing cells is an essential point for the study, so it is important to show convincing images. Co-localization is also not convincing in Figure 2B. In addition, the arrows seem to indicate cd59-positive cells and not Sox10:megfp SCs or OLs, as stated in the figure legend. Co-IF showing cd59:tagrfp expression in Sox10-positive SCs is more convincing. Please, add co-IF images at 3 or 4 dpf to illustrate quantification shown in Figure 2E. According to the Material and methods section, the IF images seem to have been taken by confocal microscopy. Please, add this information in the figure legend.

Thank you for pointing this out. In these studies, we are using a membrane-tethered transgenic line to label SCs. Therefore, we don’t expect precise co-localization of *cd59* with this transgene. Instead, we expect to see *cd59* expression within *sox10*-positive cells. We have added the missing arrows to the orange channel (Figure 2 —figure supplement 1C) and clarified that these are confocal images.

In addition, we have also added Imaris renderings to highlight the *cd59* puncta that are localized within the SCs and OLs (see Figure 2C, Figure 2 —figure supplement 1D, Figure 2 —figure supplement 2B, and Figure 3I).

We have adjusted the figure legend to make it clear that the arrows are pointing to *cd59*-postive SCs and OLs.

We have added the 3 dpf time point to the figure as well (see Figure 2E).

3. Figure 2¬—figure supplement 2: in panel A, co-localization is again not obvious at all. In panels B-D, what are we looking at? IF or in vivo labeling? Could the authors quickly clarify the expected subcellular localization of the different egfp reporters used here to avoid the need to look for this information in the referenced articles?

Thank you for your comment. We have clarified in the figure legend that these are in vivo images as well as described the expected subcellular localizations.

4. Figure 3G: same comment as above, not convincing. I really do not understand what we are looking at. In Figure 3H, the authors show a graph representing the number of puncta per cell. Does it mean that what is shown in Figure 3G is a single cell? To me, this does not look like a single cell. How do the authors define the limits of single cells? They need to show images with several cells so that we have an idea of how straightforward it is to detect the limits of each cell. In the figure legend of Figure 3G, it is not written that the images presented show single cells.

Thank you for your comment. The images do depict single cells labeled with a membrane-tethered transgene. We have added Imaris renderings to make this more clear (see Figure 2C, Figure 2 —figure supplement 1D, Figure 2 —figure supplement 2B, and Figure 3I). We have updated the figure legend to reflect this information.

5. Figure 5A: THE EM images of the mutant and the control look very different in quality. Could the authors show images of comparable quality/resolution? Looking at these images, I wonder how easy it is to count individual wraps in a reliable manner. In Figure 5D, it seems that the total number of axons has been used as n for statistics and not the average of all axons per fish with a n of 3, as stated in the figure legend. For statistical analyses, the n should be the number of fish, n=3. If the n is too high, there is a risk of finding a significant difference where there is no real difference. Please, re-calculate in case the n used was indeed the number of axons and not the number of fish.

Thank you for your comment. We have added a different representative image for the mutant (see Figure 5A). We used the total number of axons for statistics because with TEM it is difficult to section exactly at the same position along the anterior-posterior axis of the larvae. By analyzing all of the axons rather than the average, we can account for anterior-posterior differences as well as visualize the distribution the data, which would be lost by averaging per fish. We have also updated our figure legend to better clarify our statistics.

6. Figure 5C: The NF186 staining is not convincing at all. This needs to be improved. At the moment, it appears very unspecific. Co-localization with another nodal component or another antibody with a less non-specific signal could help. In addition, the resolution of the images shown in Figure 5B is not good enough and should also be improved. The mbpa:tagrfp-caax does not help much either. It would be better to use an antibody to detect MBP or Neurofilament, for example.

Thank you for your comment. Please see our previous responses to these comments.

7. Figure 6A: Again, the staining here for C5b-9/C5b-8 is not convincing. First, the signal is very low and we don't see much. Second, it is not clear whether the authors permeabilized the tissues or not (this information does not seem to be given in the Material and methods section). C5b-9/C5b-8 should be localized at the cell surface, so a cross-view should be added to the longitudinal view to demonstrate the membrane localization.

Thank you for your comment. We have added our antibody controls to the supplemental figures (please see Figure 6 —figure supplement 1A) demonstrating that we can increase MAC deposition by inducing complement activation (either through heat-related damage or DNase-elicited DNA damage). We also do not observe signal when the primary antibody is not present. Based on our controls, we do not think the extra MAC labeling is background. Rather, we believe that MAC deposition has increased globally in the *cd59* mutant embryos. This is not surprising given that complement activation leads to a positive feedback loop of more complement and immune activation, which is likely occurring in the *cd59* mutants.

To help clarify the MAC data, we have also added Imaris renderings of the MACs that are bound to the SC membranes, demonstrating that there are more MACs embedded in the *cd59* mutant SC membranes compared to wildtype SCs (see Figure 6B).

Also, in regards our methods, we did permeabilize the tissue according to the protocol we cited in the methods (Fontenas and Kucenas 2021).

8. Discussion: This part does not really discuss in depth the main claimed findings of the study and does not discuss the potential implication of the new findings in relation to the CD59 human deficiency and neurological dysfunctions resistant to treatment by complement inhibitors. This part thus appears incomplete.

Thank you for your comment. We have expanded our discussion and added our thoughts about these findings in relation to human CD59 dysfunction (please see line 642).

Reviewer #2 (Recommendations for the authors):This is an interesting and relevant manuscript and nice work, and everyone in the field should benefit from reading it. My congratulations! I have just a few observations that the authors may want to consider for their revision-As CD59 is only one out of several factors affecting Schwann cell development (GPCRs, ECM, transcription factors,.…), I feel phrasing that CD59 'orchestrates' SC development (in the title) may be a little too strong. 'Regulates', maybe?

Thank you for your comment. We have incorporated your suggestion into the title.

-Some letters and numbers are too small to be deciphered in Figure 1 Suppl 1 A, C1, C2 and D

Thank you for pointing this out. We have adjusted the text to be more easily viewed.

-Image quality is comparatively poor in Figure 1 Sup1 A, B1, B2, B3. This may be just in my PDF copy, but if not the authors may want to search for ways to improve them.

Thank you for letting us know. Our PDF copies and printed copies have good image quality, but we will ensure that if our manuscript is accepted for publication, this figure is quality checked.

-In Figure 1 Sup1 D legend, I suppose the authors mean 'myelinating OLCs', not 'mouse OLCs'

Thank you for noticing this! We have corrected the typo.

-Page 26, Line 226-227, I suppose the authors mean 117 AA, not 177 AA

Thank you for noticing this! We have corrected the typo.

- In case suitable samples and antibodies are available, Figure 3 would benefit from Western blot validation of the lack of CD59 in the zebrafish line.

Thank you for the suggestion. Alas, we tried seven different antibodies and none were suitable for Western blot or IF. This is a common issue in the zebrafish field with many antibodies.

-Figure 4 F, the authors may want to consider if these are truly representative images. The difference is not easily spotted.

Thank you for the suggestion. We have selected a different representative image to better illustrate the difference between groups (please see Figure 4F).

-Why not switch Figure 4 S2 and Figure 4 S1? It may possibly fit better with the line of arguments. Not essential, just a suggestion.

Thank you for this suggestion. We have reread our paper and prefer keeping the figures as they were originally cited in the text.

-Figure 5 A, can the authors please double-check if these are truly representative images? I am asking because the axon diameters seem to differ in the mutants as judged by the example images. Are there enough suitable samples to quantify axonal diameters? Is it possible to say if axons are normally sorted out of bundles, i.e. to quantify the numbers and diameters of axons in bundles? Sorting defects are common pathology in the PNS, at least in mammals.

Thank you for the suggestion. We quantified the area of each axon cross section (in lieu of axon diameter since these axons are not quite circular) and saw no significant differences in the size of the axons in wildtypes and mutants (see Figure 5 —figure supplement 1I). We have also added a different representative image (see Figure 5A).

- In case suitable samples and antibodies are available, Fig6 would benefit from a Western blot for the complement. Not an essential revision

Thank you for the suggestion. Given that the embryos are so small, it would be extremely difficult to isolate enough SCs for a Western blot. We would love to see future studies, perhaps in mammals, that could follow up on this suggestion.

-Related to Fig7, in case suitable samples are available it would be interesting to see TEM of myelinated axons and nodes after Dexamethasone treatment. Not an essential revision, just interesting

Thank you for the suggestion. While this would be interesting, the TEM staff that ran the electron microscopy cores have recently left their positions, and the University of Virginia has yet to replace them. Also, we do not have samples prepared that would allow us to deliver this data in a reasonable time frame. That said, this would be a great suggestion to follow up on in future studies.

- In case the required samples are available, it would be interesting to see some examination of adult WT vs mutant zebrafish. Not an essential revision, again.

Thank you for the suggestion. This is definitely an interesting idea that we would like to focus on in a future study. However, we would like the current manuscript to focus on the developmental side of this story.